# Deep Reinforcement Learning at the Edge of the Statistical Precipice

**Rishabh Agarwal**[*]
Google Research, Brain Team
MILA, Université de Montréal

**Max Schwarzer**
MILA, Université de Montréal

**Pablo Samuel Castro**
Google Research, Brain Team

**Aaron Courville**
MILA, Université de Montréal

**Marc G. Bellemare**
Google Research, Brain Team

## Abstract

Deep reinforcement learning (RL) algorithms are predominantly evaluated by comparing their relative performance on a large suite of tasks. Most published results on deep RL benchmarks compare *point estimates* of aggregate performance such as mean and median scores across tasks, ignoring the statistical uncertainty implied by the use of a finite number of training runs. Beginning with the Arcade Learning Environment (ALE), the shift towards computationally-demanding benchmarks has led to the practice of evaluating only a small number of runs per task, exacerbating the statistical uncertainty in point estimates. In this paper, we argue that reliable evaluation in the few-run deep RL regime cannot ignore the uncertainty in results without running the risk of slowing down progress in the field. We illustrate this point using a case study on the Atari 100k benchmark, where we find substantial discrepancies between conclusions drawn from point estimates alone versus a more thorough statistical analysis. With the aim of increasing the field's confidence in reported results with *a handful of runs*, we advocate for reporting interval estimates of aggregate performance and propose performance profiles to account for the variability in results, as well as present more robust and efficient aggregate metrics, such as interquartile mean scores, to achieve small uncertainty in results. Using such statistical tools, we scrutinize performance evaluations of existing algorithms on other widely used RL benchmarks including the ALE, Procgen, and the DeepMind Control Suite, again revealing discrepancies in prior comparisons. Our findings call for a change in how we evaluate performance in deep RL, for which we present a more rigorous evaluation methodology, accompanied with an open-source library *rliable*[2], to prevent unreliable results from stagnating the field.

## 1  Introduction

Research in artificial intelligence, and particularly deep reinforcement learning (RL), relies on evaluating *aggregate* performance on a diverse suite of tasks to assess progress. Quantitative evaluation on a suite of tasks, such as Atari games [5], reveals strengths and limitations of methods while simultaneously guiding researchers towards methods with promising results. Performance of RL algorithms is usually summarized with a *point estimate* of task performance measure, such as mean and median performance across tasks, aggregated over independent training runs.

A small number of training runs (Figure 1) coupled with high variability in performance of deep RL algorithms [16, 17, 41, 68, 70], often leads to substantial statistical uncertainty in reported point

---

[*]**Outstanding Paper Award**. Correspondence to Rishabh <rishabhagarwal@google.com>.
[2]https://github.com/google-research/rliable

35th Conference on Neural Information Processing Systems (NeurIPS 2021).

estimates. While evaluating more runs per task has been prescribed to reduce uncertainty and obtain reliable estimates [20, 41, 49], 3-10 runs are prevalent in deep RL as it is often computationally prohibitive to evaluate more runs. For example, 5 runs each on 50+ Atari 2600 games in ALE using standard protocol requires more than 1000 GPU training days [15]. As we move towards more challenging and complex RL benchmarks (*e.g.,* StarCraft [110]), evaluating more than a handful of runs will become increasingly demanding due to increased amount of compute and data needed to tackle such tasks. Additional confounding factors, such as exploration in the low-data regime, exacerbates the performance variability in deep RL – as seen on the Atari 100k benchmark [50] – often requiring many more runs to achieve negligible statistical uncertainty in reported estimates.

Ignoring the statistical uncertainty in deep RL results gives a false impression of fast scientific progress in the field. It inevitably evades the question: "Would similar findings be obtained with new independent runs under different random conditions?" This could steer researchers towards superficially beneficial methods [11, 12, 25], often at the expense of better methods being neglected or even rejected early [67, 74] as such methods fail to outperform inferior methods simply due to less favorable random conditions. Furthermore, only reporting point estimates obscures nuances in comparisons [85] and can erroneously lead the field to conclude which methods are *state-of-the-art* [63, 84], ensuing wasted effort when applied in practice [108]. Moreover, not report-

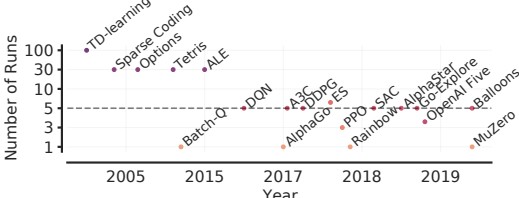

Figure 1: **Number of runs in RL over the years**. Beginning with DQN [75] on the ALE, 5 or less runs are common in the field. Here, we show representative RL papers with empirical results, in the order of their publication year: TD-learning [99], Sparse coding [100], Options [102], Tetris (CEM) [103], Batch-Q [31], ALE [5], DQN [75], AlphaGo [96], A3C [76], DDPG [62], ES [88], PPO [92], SAC [36], Rainbow [42], AlphaStar [110], Go-Explore [28], OpenAI Five [8], Balloon navigation [7] and MuZero [91].

ing the uncertainty in deep RL results makes them difficult to reproduce except under the *exact* same random conditions, which could lead to a *reproducibility crisis* similar to the one that plagues other fields [4, 44, 78]. Finally, unreliable results could erode trust in deep RL research itself [45].

In this work, we show that recent deep RL papers compare unreliable point estimates, which are dominated by statistical uncertainty, as well as exploit non-standard evaluation protocols, using a case study on Atari 100k (Section 3). Then, we illustrate how to reliably evaluate performance with only *a handful of runs* using a more rigorous evaluation methodology that accounts for uncertainty in results (Section 4). To exemplify the necessity of such methodology, we scrutinize performance evaluations of existing algorithms on widely used benchmarks, including the ALE [5] (Atari 100k, Atari 200M), Procgen [18] and DeepMind Control Suite [104], again revealing discrepancies in prior comparisons (Section 5). Our findings call for a change in how we evaluate performance in deep RL, for which we present a better methodology to prevent unreliable results from stagnating the field.

How do we reliably evaluate performance on deep RL benchmarks with only a handful of runs? As a practical solution that is easily applicable with 3-10 runs per task, we identify three statistical tools (Table 1) for improving the quality of experimental reporting. Since any performance estimate based on a finite number of runs is a *random variable*, we argue that it should be treated as such. Specifically, we argue for reporting aggregate performance measures using *interval estimates* via stratified bootstrap confidence intervals, as opposed to point estimates. Among prevalent aggregate measures, mean can be easily dominated by performance on a few outlier tasks, while median has high variability and zero performance on nearly half of the tasks does not change it. To address these deficiencies, we present more *efficient* and *robust* alternatives, such as *interquartile mean*, which are not unduly affected by outliers and have small uncertainty even with a handful of runs. Furthermore, to reveal the variability in performance across tasks, we propose reporting performance distributions across all runs. Compared to prior work [5, 83], these distributions result in *performance profiles* [26] that are statistically unbiased, more robust to outliers, and require fewer runs for smaller uncertainty.

## 2 Formalism

We consider the setting in which a reinforcement learning algorithm is evaluated on $M$ tasks. For each of these tasks, we perform $N$ independent runs[3] which each provide a scalar, *normalized score*

---

[3]A run can be different from using a fixed random seed. Indeed, fixing the seed may not be able to control all sources of randomness such as non-determinism of ML frameworks with GPUs (*e.g.,* Figure A.13).

Table 1: Our recommendations for reliable evaluation, easily applicable with a handful of runs. Refer to Section 4 for details about recommendations and Section 5 for their application to widely-used RL benchmarks.

| Desideratum | Current Evaluation Protocol | Our Recommendation |
|---|---|---|
| Uncertainty in aggregate performance | **Point estimates**
• Ignore statistical uncertainty
• Hinder *results reproducibility* | Interval estimates via **stratified bootstrap confidence intervals** |
| Variability in performance across tasks and runs | **Tables with mean scores per task**
• Overwhelming beyond a few tasks
• Standard deviations often omitted
• Incomplete picture for multimodal and heavy-tailed distributions | **Performance profiles** (*score distributions*)
• Show tail distribution of scores on combined runs across tasks
• Allow qualitative comparisons
• Easily read any score percentile |
| Aggregate metrics for summarizing performance across tasks | **Mean**
• Often dominated by performance on outlier tasks
**Median**
• Requires large number of runs to claim improvements
• Poor indicator of overall performance: zero scores on nearly half the tasks do not affect it | **Interquartile Mean** (IQM) across all runs
• Performance on middle 50% of combined runs
• Robust to outlier scores but more statistically efficient than median
To show other aspects of performance gains, report average *probability of improvement* and *optimality gap*. |

$x_{m,n}$, $m = 1, \ldots, M$ and $n = 1, \ldots, N$. These normalized scores are obtained by linearly rescaling per-task scores[4] based on two reference points; for example, performance on the Atari games is typically normalized with respect to a random agent and an average human, who are assigned a normalized score of 0 and 1 respectively [75]. We denote the set of normalized scores by $x_{1:M,1:N}$.

In most experiments, there is inherent randomness in the scores obtained from different runs. This randomness can arise from stochasticity in the task, exploratory choices made during learning, randomized initial parameters, but also software and hardware considerations such as non-determinism in GPUs and in machine learning frameworks [116]. Thus, we model the algorithm's normalized score on the $m^{th}$ task as a real-valued random variable $X_m$. Then, the score $x_{m,n}$ is a realization of the random variable $X_{m,n}$, which is identically distributed as $X_m$. For $\tau \in \mathbb{R}$, we define the tail distribution function of $X_m$ as $F_m(\tau) = \mathrm{P}(X_m > \tau)$. For any collection of scores $y_{1:K}$, the *empirical tail distribution function* is given by $\hat{F}(\tau; y_{1:K}) = \frac{1}{K} \sum_{k=1}^{K} \mathbb{1}[y_k > \tau]$. In particular, we write $\hat{F}_m(\tau) = \hat{F}(\tau; x_{m,1:N})$.

The *aggregate performance* of an algorithm maps the set of normalized scores $x_{1:M,1:N}$ to a scalar value. Two prevalent aggregate performance metrics are the mean and median normalized scores. If we denote by $\bar{x}_m = \frac{1}{N} \sum_{n=1}^{N} x_{m,n}$ the average score on task $m$ across $N$ runs, then these aggregate metrics are $\mathrm{Mean}(\bar{x}_{1:M})$ and $\mathrm{Median}(\bar{x}_{1:M})$. More precisely, we call these *sample mean* and *sample median* over the task means since they are computed from a finite set of $N$ runs. Since $\bar{x}_m$ is a realization of the random variable $\bar{X}_m = \frac{1}{N} \sum_{n=1}^{N} X_{m,n}$, the sample mean and median scores are *point estimates* of the random variables $\mathrm{Mean}(\bar{X}_{1:M})$ and $\mathrm{Median}(\bar{X}_{1:M})$ respectively. We call *true mean* and *true median* the metrics that would be obtained if we had unlimited experimental capacity ($N \to \infty$), given by $\mathrm{Mean}(\mathbb{E}[X_{1:M}])$ and $\mathrm{Median}(\mathbb{E}[X_{1:M}])$ respectively.

**Confidence intervals** (CIs) for a finite-sample score can be interpreted as an estimate of plausible values for the true score. A $\alpha \times 100\%$ CI computes an interval such that if we rerun the experiment and construct the CI using a different set of runs, the fraction of calculated CIs (which would differ for each set of runs) that contain the true score would tend towards $\alpha \times 100\%$, where $\alpha \in [0, 1]$ is the nominal coverage rate. 95% CIs are typically used in practice. If the true score lies outside the 95% CI, then a sampling event has occurred which had a probability of 5% of happening by chance.

---

[4]Often the average undiscounted return obtained during an episode (see Sutton and Barto [101] for an explanation of the reinforcement learning setting).

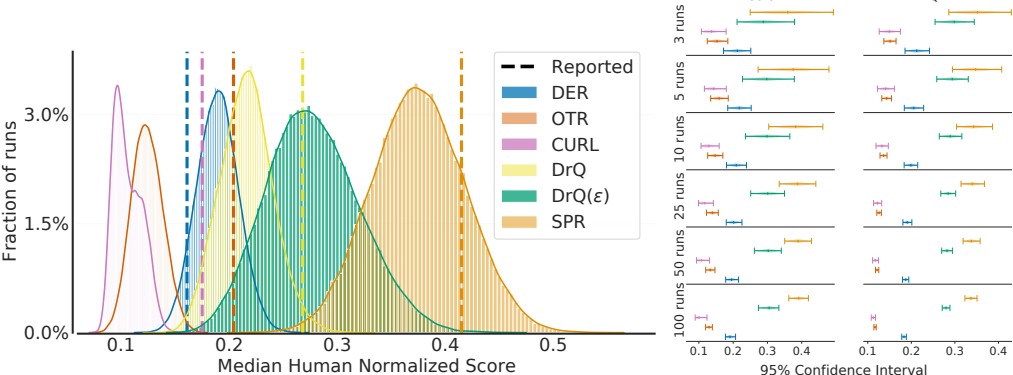

Figure 2: **Left**. **Distribution of median normalized scores** computed using 100,000 different sets of $N$ runs subsampled uniformly with replacement from 100 runs. For a given algorithm, the sampling distribution shows the variation in the median scores when re-estimated using a different set of runs. The reported *point estimates* of median in publications, as shown by dashed lines, do not provide any information about the variability in median scores and severely overestimate or underestimate the expected median. We use the same number of runs as reported by publications: $N = 5$ runs for DER, OTR and DRQ, $N = 10$ runs for SPR and $N = 20$ runs for CURL. **Right**. **95% CIs** for median and IQM scores (Section 4.3) for varying $N$. There is a substantial uncertainty in median scores even with 50 runs. IQM has much smaller CIs than median. Note that when CIs overlap, properly accounting for uncertainty entails computing CIs for score differences (Figure A.15).

**Remark**. Following Amrhein et al. [2], Romer [87], Wasserstein et al. [112], we recommend using confidence intervals for measuring the uncertainty in results and showing effect sizes (*e.g.,* performance improvements over baseline) that are compatible with the given data. Furthermore, we emphasize using statistical thinking but avoid statistical significance tests (*e.g.,* $p$-value $< 0.05$) because of their dichotomous nature (significant *vs.* not significant) and common misinterpretations [33, 35, 73] such as 1) lack of statistically significant results does not demonstrate the absence of effect (Figure 2, right), and 2) given enough data, any trivial effect can be statistically significant but may not be practically significant.

## 3   Case Study: The Atari 100k benchmark

We begin with a case study to illustrate the pitfalls arising from the naïve use of point estimates in the few-run regime. Our case study concerns the Atari 100k benchmark [50], an offshoot of the ALE for evaluating data-efficiency in deep RL. In this benchmark, algorithms are evaluated on only 100k steps (2-3 hours of game-play) for each of its 26 games, versus 200M frames in the ALE benchmark. Prior reported results on this benchmark have been computed mostly from 3 [39, 55, 59, 72, 89, 95] or 5 runs [50, 51, 53, 54, 64, 66, 86, 107, 115], and more rarely, 10 [65, 93] or 20 runs [56].

Our case study compares the performance of five recent deep RL algorithms, namely: (1) DER [107] and (2) OTR [51], (3) DRQ[5] [53], (4) CURL [56], and (5) SPR [93]. We chose these methods as representative of influential algorithms within this benchmark. Since good performance on one game can result in unduly high sample means without providing much information about performance on other games, it is common to measure performance on Atari 100k using sample medians. Refer to Appendix A.2 for more details about the experimental setup.

We investigate statistical variations in the few-run regime by evaluating 100 independent runs for each algorithm, where the score for a run is the average returns obtained in 100 evaluation episodes taking place after training. Each run corresponds to training one algorithm on each of the 26 games in Atari 100k. This provides us with $26 \times 100$ scores per algorithm, which we then subsample with replacement to 3–100 runs. The subsampled scores are then used to produce a collection of point estimates whose statistical variability can be measured. We begin by using this experimental protocol to highlight statistical concerns regarding median normalized scores.

**High variability in reported results.** Our first observation is that the sample medians reported in the literature exhibit substantial variability when viewed as random quantities that depend on a

---

[5]DRQ codebase uses non-standard evaluation hyperparameters. Instead, DRQ($\varepsilon$) corresponds to DRQ with standard $\varepsilon$-greedy parameters [14, Table 1] in ALE. See Appendix for more details.

small number of sample runs (Figure 2, left). This shows that there is a fairly large potential for drawing erroneous conclusions based on point estimates alone. As a concrete example, our analysis suggests that DER may in fact be better than OTR, unlike what the reported point estimates suggest. We conclude that in the few-run regime, point estimates are unlikely to provide definitive answers to the question: "Would we draw the same conclusions were we to re-evaluate our algorithm with a different set of runs?"

**Substantial bias in sample medians**. The sample median is a biased estimator of the true median: $\mathbb{E}[\text{Median}(\bar{X}_{1:M})] \neq \text{Median}(\mathbb{E}[X_{1:M}])$ in general. In the few-run regime, we find that this bias can dominate the comparison between algorithms, as evidenced in Figure 3. For example, the score difference between sample medians with 5 and 100 runs for SPR (+0.03 points) is about 36% of its mean improvement over DrQ($\varepsilon$) (+0.08 points). Adding to the issue, the magnitude and sign of this bias strongly depends on the algorithm being evaluated.

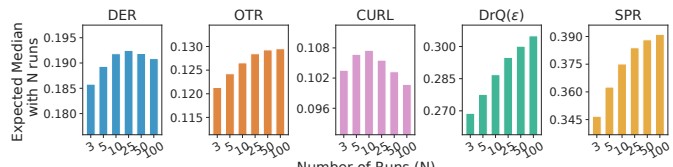

Figure 3: **Expected sample median** of task means. The expected score for $N$ runs is computed by repeatedly subsampling $N$ runs with replacement out of 100 runs for 100,000 times.

**Statistical concerns cannot be satisfactorily addressed with few runs.** While claiming improvements with 3 or fewer runs may naturally raise eyebrows, folk wisdom in experimental RL suggests that 20 or 30 runs are enough. By calculating 95% confidence interval[6] on sample medians for a varying number of runs (Figure 2, right), we find that this number is closer to 50–100 runs in Atari 100k – far too many to be computationally feasible for most research projects.

Consider a setting in which an algorithm is known to be better – what is the reliability of median and IQM (Section 4.3) for accurately assessing performance differences as the number of runs varies? Specifically, we consider two identical $N$-run experiments involving SPR, except that we artificially inflate one of the experiments' scores by a fixed fraction or *lift* of $+\ell\%$ (Figure 4). In particular, $\ell = 0$ corresponds to running the same experiment twice but with different runs. We find that statistically defensible improvements with median scores is only achieved for 25 runs ($\ell = 25$) and 100 runs ($\ell = 10$). With $\ell = 0$, even 100 runs are insufficient, with deviations of 20% possible.

**Changes in evaluation protocols invalidates comparisons to prior work.** A typical and relatively safe approach for measuring the performance of an RL algorithm is to average the scores received in their final training episodes [69]. However, the field has seen a number of alternative protocols used, including reporting the maximum evaluation score achieved during training [1, 3, 75] or across multiple runs [32, 47, 82]. A similar protocol is also used by CURL and SUNRISE [59] (Appendix A.4).

Results produced under alternative protocols involving maximum are generally incomparable with end-performance reported results. On Atari 100k, we find that the two protocols produce substantially different results (Figure 5), of a magnitude greater than the actual difference in score. In particular, evaluating DER with CURL's protocol results in scores far above those reported for CURL. In other words, this gap in evaluation procedures resulted in CURL being assessed as achieving a greater true median than DER, where our experiment gives strong support to DER being superior. Similarly, we find that a lot of SUNRISE's improvement over DER can be explained by the change in evaluation protocol (Figure 5). Refer to Appendix A.4 for discussion on pitfalls of such alternative protocols.

# 4 Recommendations and Tools for Reliable Evaluation

Our case study shows that the increase in the number of runs required to address the statistical uncertainty issues is typically infeasible for computationally demanding deep RL benchmarks. In this section, we identify three tools for improving the quality of experimental reporting in the few-run regime, all aligned with the principle of accounting for statistical uncertainty in results.

## 4.1 Stratified Bootstrap Confidence Intervals

We first reaffirm the importance of reporting interval estimates to indicate the range within which an algorithm's aggregate performance is believed to lie. Concretely, we propose using bootstrap CIs [29]

---

[6]Specifically, we use the $m/n$ bootstrap [9] to calculate the interval between $[2.5^{th}, 97.5^{th}]$ percentiles of the distribution of sample medians (95% CIs).

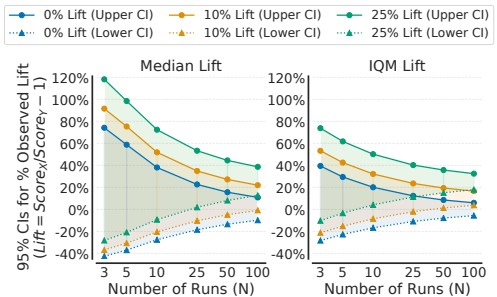

Figure 4: **Detecting score lifts**. **Left**. 95% CIs for observed lift with median scores, and **Right**. 95% CIs for observed lift with IQM (Section 4.3) when comparing SPR with an algorithm that performs $\ell\%$ better. IQM requires fewer runs than median for small uncertainty.

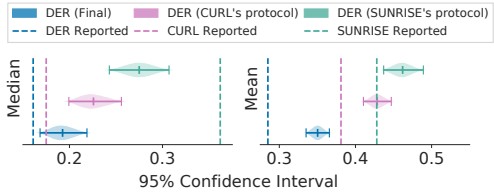

Figure 5: **Normalized DER scores** with non-standard evaluation protocols. Gains from SUNRISE and CURL over DER can mostly be explained by such protocols.

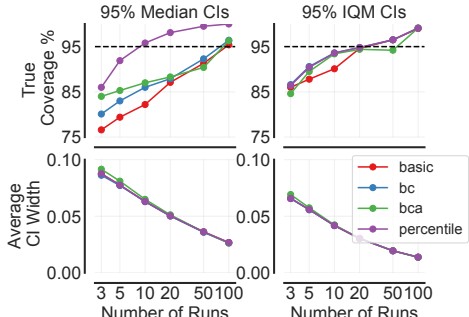

Figure 6: **Validating 95% Stratified Bootstrap CIs** for a varying number of runs for median and IQM scores for DER. The true coverage % is computed by sampling 10,000 sets of K runs without replacement from 200 runs and checking the fraction of 95% CIs that contains the true estimate approximation based on 200 runs. Note that we evaluate additional 100 runs for DER for an accurate point estimate. Percentile CIs has the best coverage while achieving a small width compared to other methods. Also, CI widths for IQM are much smaller than that of median. We also note that with 3 runs, bootstrap CIs underestimate the true 95% CIs and might require a larger nominal coverage rate to achieve true 95% coverage.

with stratified sampling for aggregate performance, a method that can be applied to small sample sizes and is better justified than reporting sample standard deviations in this context. While prior work has recommended using bootstrap CIs for reporting uncertainty in single task mean scores with $N$ runs [16, 20, 41], this is less useful when $N$ is small (Figure A.18), as *bootstrapping* assumes that re-sampling from the data approximates sampling from the true distribution. We can do better by aggregating samples across tasks, for a total of $MN$ random samples.

To compute the stratified bootstrap CIs, we re-sample runs with replacement independently for each task to construct an empirical bootstrap sample with $N$ runs each for $M$ tasks from which we calculate a statistic and repeat this process many times to approximate the sampling distribution of the statistic. We measure the reliability of this technique in Atari 100k for variable $N$, by comparing the nominal coverage of 95% to the "true" coverage from the estimated CIs (Figure 6) for different bootstrap methods (see [30] and Appendix A.5). We find that percentile CIs provide good interval estimates for as few as $N = 10$ runs for both median and IQM scores (Section 4.3).

## 4.2 Performance Profiles

Most deep RL benchmarks yield scores that vary widely between tasks and may be heavy-tailed, multimodal, or possess outliers (*e.g.,* Figure A.14). In this regime, both point estimates, such as mean and median scores, and interval estimates of these quantities paint an incomplete picture of an algorithm's performance [24, Section 3]. Instead, we recommend the use of *performance profiles* [26], commonly used in benchmarking optimization software. While performance profiles from Dolan and Moré [26] correspond to empirical cumulative distribution functions without any uncertainty estimates, profiles proposed herein visualize the empirical tail distribution function (Section 2) of a random score (higher curve is better), with pointwise confidence bands based on stratified bootstrap.

By representing the entire set of normalized scores $x_{1:M,1:N}$ visually, performance profiles reveal performance variability across tasks much better than interval estimates of aggregate metrics. Although tables containing per-task mean scores and standard deviations can reveal this variability, such tables tend to be overwhelming for more than a few tasks.[7] In addition, performance profiles are robust to outlier runs and insensitive to small changes in performance across all tasks [26].

In this paper, we propose the use of a performance profile we call run-score distributions or simply *score distributions* (Figure 7, right), particularly well-suited to the few-run regime. A score distribution shows the fraction of runs above a certain normalized score and is given by

---

[7]In addition, standard deviations are sometimes omitted from tables due to space constraints.

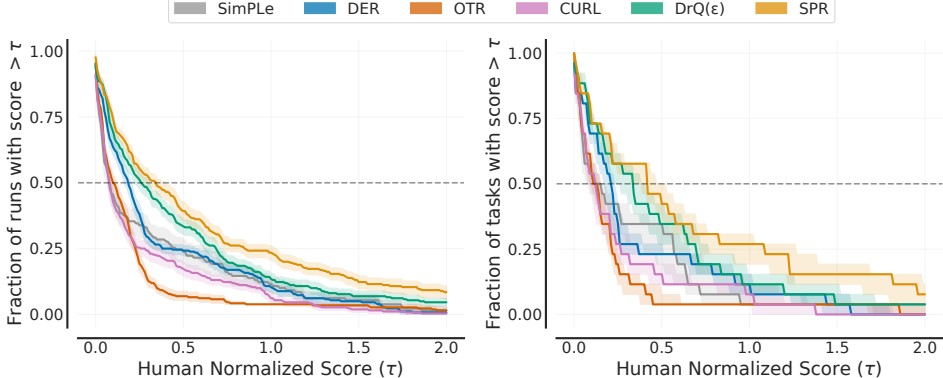

Figure 7: **Performance profiles on Atari 100k** based on score distributions (**left**), which we recommend, and average score distributions (**right**). Shaded regions show pointwise 95% confidence bands based on percentile bootstrap with stratified sampling. The profiles on the left are more robust to outliers and have smaller confidence bands. We use 10 runs to show the robustness of profiles with a few runs. For SimPLe [50], we use the 5 runs from their reported results. The $\tau$ value where the profiles intersect $y = 0.5$ shows the median while for a non-negative random variable, area under the performance profile corresponds to the mean.

$$\hat{F}_X(\tau) = \hat{F}(\tau; x_{1:M,1:N}) = \frac{1}{M} \sum_{m=1}^{M} \hat{F}_m(\tau) = \frac{1}{M} \sum_{m=1}^{M} \frac{1}{N} \sum_{n=1}^{N} \mathbb{1}[x_{m,n} > \tau]. \tag{1}$$

One advantage of the score distribution is that it is an unbiased estimator of the underlying distribution $F(\tau) = \frac{1}{N} \sum_{m=1}^{M} F_m(\tau)$. Another advantage is that an outlier run with extremely high score can change the output of score distribution for any $\tau$ by at most a value of $\frac{1}{MN}$.

It is useful to contrast score distributions to average-score distributions, originally proposed in the context of the ALE [5] as a generalization of the median score. Average-score distributions correspond to the performance profile of a random variable $\bar{X}$, $\hat{F}_{\bar{X}}(\tau) = \hat{F}(\tau; \bar{x}_{1:M})$, which shows the fraction of tasks on which an algorithm performs better than a certain score. However, such distributions are a biased estimate of the thing they seek to represent. Run-score distributions are more robust than average-score distributions, as they are a step function in $1/MN$ versus $1/M$ intervals, and typically has less variance: $\sigma_X^2 = \frac{1}{M^2N} \sum_{m=1}^{M} F_m(\tau)(1 - F_m(\tau))$ versus $\sigma_{\bar{X}}^2 = \frac{1}{M^2} \sum_{m=1}^{M} F_{\bar{X}_m}(\tau)(1 - F_{\bar{X}_m}(\tau))$. Figure 7 illustrates these differences.

### 4.3 Robust and Efficient Aggregate Metrics

Performance profiles allow us to compare different methods at a glance. If one curve is strictly above another, the better method is said to *stochastically dominate*[8] the other [27, 61]. In RL benchmarks with a large number of tasks, however, stochastic dominance is rarely observed: performance profiles often intersect at multiple points. Finer quantitative comparisons must therefore entail aggregate metrics.

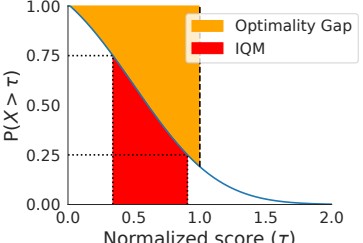

Figure 8: **Aggregate metrics**. For a non-negative random variable $X$, IQM corresponds to the red shaded region while optimality gap corresponds to the orange shaded region in the performance profile of $X$.

We can extract a number of aggregate metrics from score distributions, including median (mixing runs and tasks) and mean normalized scores (matching our usual definition). As we already argued that these metrics are deficient, we now consider interesting alternatives also derived from score distributions.

As an alternative to median, we recommend using the **interquartile mean** (IQM). Also called 25% trimmed mean, IQM discards the bottom and top 25% of the runs and calculates the mean score of the remaining 50% runs ($= \lfloor NM/2 \rfloor$ for $N$ runs each on $M$ tasks). IQM interpolates between mean and median across runs, which are 0% and almost 50% trimmed means

---

[8]A random variable $X$ has stochastic dominance over random variable $Y$ if $P(X > \tau) \geq P(Y > \tau)$ for all $\tau$, and for some $\tau$, $P(X > \tau) > P(Y > \tau)$.

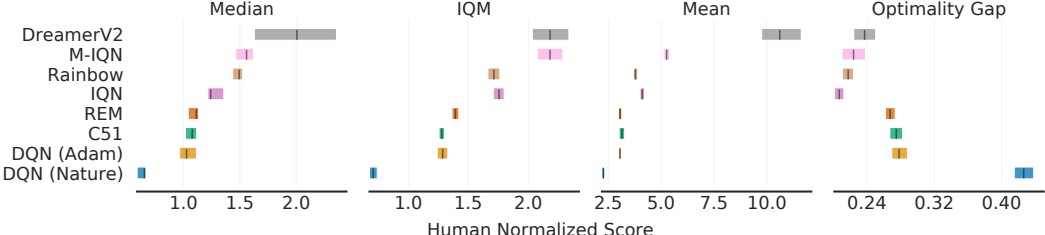

Figure 9: **Aggregate metrics on Atari 200M** with 95% CIs based on 55 games with sticky actions [69]. Higher mean, median and IQM scores and lower optimality gap are better. The CIs are estimated using the percentile bootstrap with stratified sampling. IQM typically results in smaller CIs than median scores. Large values of mean scores relative to median and IQM indicate being dominated by a few high performing tasks, for example, DreamerV2 and M-IQN obtain normalized scores above 50 on the game JAMESBOND. Optimality gap is less susceptible to outliers compared to mean scores. We compare DQN (Nature) [75], DQN with Adam optimizer, C51 [6], REM [1], Rainbow [42], IQN [22], Munchausen-IQN (M-IQN) [109], and DreamerV2 [38]. All results are based on 5 runs per game except for M-IQN and DreamerV2 which report results with 3 and 11 runs.

respectively. Compared to sample median, IQM is a better indicator of overall performance as it is calculated using 50% of the combined runs while median only depends on the performance ordering across tasks and not on the magnitude except at most 2 tasks. For example, zero scores on nearly half of the tasks does not affect the median while IQM exhibits a severe degradation. Compared to mean, IQM is robust to outliers, yet has considerably less bias than median (Figure A.17). While median is more robust to outliers than IQM, this robustness comes at the expense of statistical efficiency, which is crucial in the few-run regime: IQM results in much smaller CIs (Figure 2 (right) and 6) and is able to detect a given improvement with far fewer runs (Figures 4 and A.15).

As a robust alternative to mean, we recommend using the **optimality gap**: the amount by which the algorithm fails to meet a minimum score of $\gamma = 1.0$ (orange region in Figure 8). This assumes that a score of 1.0 is a desirable target beyond which improvements are not very important, for example when the aim is to obtain human-level performance [*e.g.,* 3, 23]. Naturally, the threshold $\gamma$ may be chosen differently, which we discuss further in Appendix A.7.

If one is interested in knowing how robust an improvement from an algorithm $X$ over an algorithm $Y$ is, another possible metric to consider is the average **probability of improvement** – this metric shows how likely it is for $X$ to outperform $Y$ on a randomly selected task. Specifically, $P(X > Y) = \frac{1}{M} \sum_{m=1}^{M} P(X_m > Y_m)$, where $P(X_m > Y_m)$ (Equation A.2) is the probability that $X$ is better than $Y$ on task $m$. Note that, unlike IQM and optimality gap, this metric does not account for the size of improvement. While finding the best aggregate metric is still an open question and is often dependent on underlying normalized score distribution, our proposed alternatives avoid the failure modes of prevalent metrics while being robust and requiring fewer runs to reduce uncertainty.

## 5   Re-evaluating Evaluation on Deep RL Benchmarks

**Arcade Learning Environment**. Training RL agents for 200M frames on the ALE [5, 69] is the most widely recognized benchmark in deep RL. We revisit some popular methods which demonstrated progress on this benchmark and reveal discrepancies in their findings as a consequence of ignoring the uncertainty in their results (Figure 9). For example, DreamerV2 [38] exhibits a large amount of uncertainty in aggregate scores. While M-IQN [109] claimed better performance than Dopamine Rainbow[9] [42] in terms of median normalized scores, their interval estimates strikingly overlap. Similarly, while C51 [5] is considered substantially better than DQN [75], the interval estimates as well as performance profiles for DQN (Adam) and C51 overlap significantly.

Figure 9 reveals an interesting limitation of aggregate metrics: depending on the choice of metric, the ordering between algorithms changes (*e.g.,* Median *vs.* IQM). The inconsistency in ranking across aggregate metrics arises from the fact that such metrics only capture a specific aspect of overall performance across tasks and runs. Additionally, the change of algorithm ranking between optimality gap and IQM/median scores reveal that while recent algorithms typically show performance gains relative to humans on average, their performance seems to be worse on games below human

---

[9]Dopamine Rainbow differs from that of Hessel et al. [42] by not including double DQN, dueling architecture and noisy networks. Also, results in [42] were reported using a single run without sticky actions.

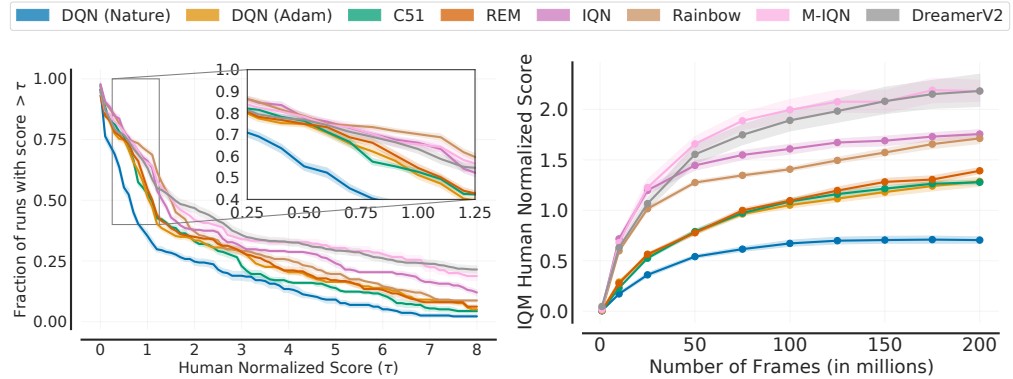

Figure 10: **Atari 200M evaluation**. **Left**. Score distributions using human-normalized scores obtained after training for 200M frames. **Right**. Sample-efficiency of agents as a function of number of frames measured via IQM human-normalized scores. Shaded regions show pointwise 95% percentile stratified bootstrap CIs.

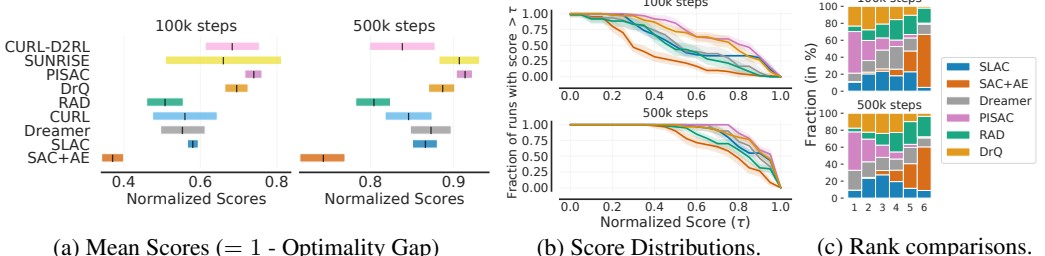

(a) Mean Scores (= 1 - Optimality Gap)  (b) Score Distributions.  (c) Rank comparisons.

Figure 11: **DeepMind Control Suite evaluation** results, averaged across 6 tasks, on the 100k and 500k benchmark. We compare SAC+AE [114], SLAC [58], Dreamer [37], CURL [98], RAD [57], DrQ [53], PISAC [60], SUNRISE [59], and CURL-D2RL [97]. The **ordering** of the algorithms in the left figure is based on their claimed relative performance – all algorithms except Dreamer claimed improvement over at least one algorithm placed below them. **(a)** Interval estimates show 95% stratified bootstrap CIs for methods with individual runs provided by their respective authors and 95% studentized CIs for CURL, CURL-D2RL, and SUNRISE. Normalized scores are computed by dividing by the maximum score (=1000). **(b)** Score distributions. **(c)** The $i^{th}$ column in the rank distribution plots show the probability that a given method is assigned rank $i$, averaged across all tasks. The ranks are estimated using 200,000 stratified bootstrap re-samples.

performance. Since performance profiles capture the full picture, they would often illustrate why such inconsistencies exist. For example, optimality gap and IQM can be both read as areas in the profile (Figure 8). The performance profile in Figure 10 (left) illustrates the nuances present when comparing different algorithms. For example, IQN seems to be better than Rainbow for $\tau \geq 2$, but worse for $\tau < 2$. Similarly, the profiles of DreamerV2 and M-IQN for $\tau < 8$ intersect at multiple points. To compare sample efficiency of the agents, we also present their IQM scores as a function of number of frames in Figure 10 (right).

**DeepMind Control Suite**. Recent continuous control papers benchmark performance on 6 tasks in DM Control [104] at 100k and 500k steps. Typically, such papers claim improvement based on higher mean scores per task regardless of the variability in those scores. However, we find that when accounting for uncertainty in results, most algorithms do not consistently rank above algorithms they claimed to improve upon (Figure 11c and 11b). Furthermore, there are huge overlaps in 95% CIs of mean normalized scores for most algorithms (Figure 11a). These findings suggest that a lot of the reported improvements are spurious, resulting from randomness in the experimental protocol.

**Procgen benchmark**. Procgen [18] is a popular benchmark, consisting of 16 diverse tasks, for evaluating generalization in RL. Recent papers report mean PPO-normalized scores on this benchmark to emphasize the gains relative to PPO [92] as most methods are built on top of it. However, Figure 12 (left) shows that PPO-normalized scores typically have a heavy-tailed distribution making the mean scores highly dependent on performance on a small fraction of tasks. Instead, we recommend using normalization based on the estimated minimum and maximum scores on ProcGen [18] and reporting aggregate metrics based on such scores (Figure A.32). While publications sometimes make binary claims about whether they improve over prior methods, such improvements are inherently probabilistic. To reveal this discrepancy, we investigate the following question: "What is the

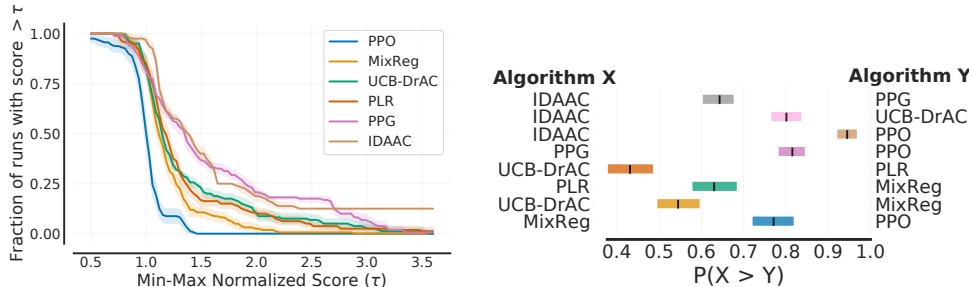

Figure 12: **Procgen evaluation** results based on easy mode comparisons [80] with 16 tasks. **Left**. Score distributions which compare PPO [92], MixReg [111], UCB-DrAC [81], PLR [48], PPG [19] and IDAAC [80]. Shaded regions indicate 95% percentile stratified bootstrap CIs. **Right**. Each row shows the probability of improvement, with 95% bootstrap CIs, that the algorithm $X$ on the left outperforms algorithm $Y$ on the right, given that $X$ was claimed to be better than $Y$. For all algorithms, results are based on 10 runs per task.

probability that an algorithm which claimed improvement over a prior algorithm performs better than it?" (Figure 12, right). While this probability does not distinguish between two algorithms which uniformly improve on all tasks by 1% and 100%, it does highlight how likely an improvement is. For example, there is only a $40 - 50\%$ chance that UCB-DrAC [81] improves upon PLR [48]. We note that a number of improvements reported in the existing literature are only $50 - 70\%$ likely.

## 6  Discussion

We saw, both in our case study on the Atari 100k benchmark and with our analysis of other widely-used RL benchmarks, that statistical issues can have a sizeable influence on reported results, in particular when point estimates are used or evaluation protocols are not kept constant within comparisons. Despite earlier calls for more experimental rigor in deep RL [16, 20, 21, 41, 49, 83] (discussed in Appendix A.3), our analysis shows that the field has not yet found sure footing in this regards.

In part, this is because the issue of reproducibility is a complex one; where our work is concerned with our confidence about and interpretation of reported results (what Goodman et al. [34] calls *results reproducibility*), others [79] have highlighted that there might be missing information about the experiments themselves (*methods reproducibility*). We remark that the problem is not solved by fixing random seeds, as has sometimes been proposed [52, 77], since it does not really address the question of whether an algorithm would perform well under similar conditions but with different seeds. Furthermore, fixed seeds might benefit certain algorithms more than others. Nor can the problem be solved by the use of dichotomous statistical significance tests, as discussed in Section 2.

One way to minimize the risks associated with statistical effects is to report results in a more complete fashion, paying close attention to bias and uncertainty within these estimates. To this end, our recommendations are summarized in Table 1. To further support RL researchers in this endeavour, we released an easy-to-use Python library, `rliable` along with a Colab notebook for implementing our recommendations, as well as all the individual runs used in our experiments[10]. Again, we emphasize the importance of published papers providing results for all runs to allow for future statistical analyses.

A barrier to adoption of evaluation protocols proposed in this work, and more generally, rigorous evaluation, is whether there are clear incentives for researchers to do so, as more rigor generally entails more nuanced and tempered claims. Arguably, doing good and reproducible science is one such incentive. We hope that our findings about erroneous conclusions in published papers would encourage researchers to avoid fooling themselves, even if that requires tempered claims. That said, a more pragmatic incentive would be if conferences and reviewers required more rigorous evaluation for publication, *e.g.,* NeurIPS 2021 checklist asks whether error bars are reported. Moving towards reliable evaluation is an ongoing process and we believe that this paper would greatly benefit it.

Given the substantial influence of statistical considerations in experiments involving 40-year old Atari 2600 video games and low-DOF robotic simulations, we argue that it is unlikely that an increase in available computation will resolve the problem for the future generation of RL benchmarks. Instead, just as a well-prepared rock-climber can skirt the edge of the steepest precipices, it seems likely that ongoing progress in reinforcement learning will require greater experimental discipline.

---

[10]Colab: `bit.ly/statistical_precipice_colab`. Individual runs: `gs://rl-benchmark-data`.

## Societal Impacts

This paper calls for statistical sophistication in deep RL research by accounting for statistical uncertainty in reported results. However, statistical sophistication can introduce new forms of statistical abuses and monitoring the literature for such abuses should be an ongoing priority for the research community. Moving towards reliable evaluation and reproducible research is an ongoing process and this paper only partly addresses it by providing tools for more reliable evaluation. That said, while accounting for uncertainty in results is not a panacea, it provides a strong foundation for trustworthy results on which the community can build upon, with increased confidence. In terms of broader societal impact of this work, we do not see any foreseeable strongly negative impacts. However, this paper could positively impact society by constituting a step forwards in rigorous few-run evaluation regime, which reduces computational burden on researchers and is "greener" than evaluating a large number of runs.

## Acknowledgments

We thank Xavier Bouthillier, Dumitru Erhan, Marlos C. Machado, David Ha, Fabio Viola, Fernando Diaz, Stephanie Chan, Jacob Buckman, Danijar Hafner and anonymous NeurIPS' reviewers for providing valuable feedback for an earlier draft of this work. We also acknowledge Matteo Hessel, David Silver, Tom Schaul, Csaba Szepesvári, Hado van Hasselt, Rosanne Liu, Simon Kornblith, Aviral Kumar, George Tucker, Kevin Murphy, Ankit Anand, Aravind Srinivas, Matthew Botvinick, Clare Lyle, Kimin Lee, Misha Laskin, Ankesh Anand, Joelle Pineau and Braham Synder for helpful discussions. We also thank all the authors who provided individual runs for their corresponding publications. We are also grateful for general support from Google Research teams in Montréal and elsewhere.

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
