# A  Appendix

## A.1  Open-source notebook and data

**Colab notebook** for producing and analyzing performance profiles, robust aggregate metrics, and interval estimates based on stratified bootstrap CIs, as well as replicating the results in the paper can be found at `bit.ly/statistical_precipice_colab`.

**Individual runs for Atari 100k**. We released the 100 runs per game for each of the 6 algorithms in the case study in a public cloud bucket at `gs://rl-benchmark-data/atari_100k`.

**Individual runs for ALE, Procgen and DM Control**. For ALE, we used the individual runs from Dopamine [14] baselines except for DreamerV2 [38], REM [1] and M-IQN [109], for which the individual run scores were obtained from the corresponding authors. We release all the individual run scores as well as final scores for ALE at `gs://rl-benchmark-data/ALE`. The Procgen results were obtained from the authors of IDAAC [80] and MixReg [48] and are released at `gs://rl-benchmark-data/procgen`. For DM Control[11], all the runs were obtained from the corresponding authors and are released at `gs://rl-benchmark-data/dm_control`.

See `agarwl.github.io/rliable` for a website for the paper.

## A.2  Atari 100k: Additional Details and Results

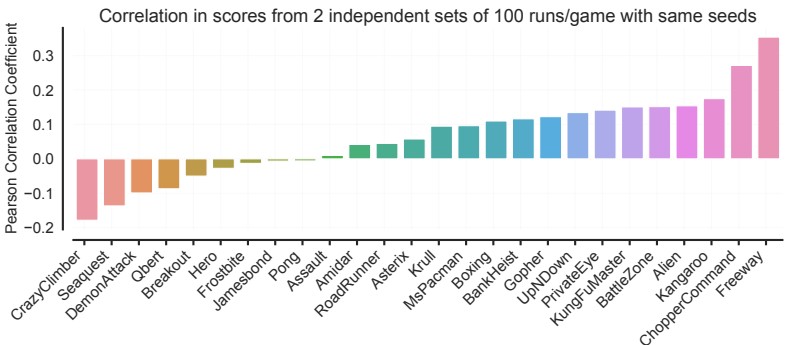

Figure A.13: **Runs can be different from using fixed random seeds**. We find that correlation between two sets of 100 runs of DER on Atari 100k using the same set of random seeds, that is, using a fixed random seed per run for Python, NumPy and JAX, is quite small. Small values of correlation coefficient highlight that fixing seeds does not ensure deterministic results due to non-determinism in GPUs. Similarly, setting random seed in PyTorch ensures reproducibility only on the same hardware.

**Code**. Due to unavailability of open-source code for DER, and OTR for Atari 100k, we re-implemented these algorithms using Dopamine [14], a reproducible deep RL framework. For CURL and SPR, we used the open-source code released by the authors while for DrQ, we used the source-code obtained from the authors. Our code for Atari 100k experiments is open-sourced as part of the Dopamine library under the `labs/atari_100k` folder. We also released a JAX [13] implementation of the full Rainbow [42] in Dopamine.

**Hyperparameters**. All algorithms build upon the Rainbow [42] architecture and we use the exact same hyperparameters specified in the corresponding publication unless specified otherwise. Akin to DrQ and SPR, we used $n$-step returns with $n = 10$ instead of $n = 20$ for DER. DrQ codebase uses non-standard evaluation hyperparameters, such as a 5% probability of selecting random actions during evaluation ($\varepsilon$-`eval`$= 0.05$). DrQ($\varepsilon$) differs DrQ in terms of using standard $\varepsilon$-greedy parameters [14, Table 1] including training $\varepsilon$ decayed to 0.01 rather than 0.1 and evaluation $\varepsilon$ set to 0.001 instead of 0.05. Refer to the gin configurations in `labs/atari_100k/configs` for more details.

---

[11]Dreamer [37] results on DM control, obtained from the corresponding author, are based on hyperparameters tuned for sample-efficiency. Compared to the original paper [37], the actor-critic learning rates were increased to $3e - 4$, the amount of free bits to 1.5, the training frequency, and the amount of pre-training to 1k steps on 10k randomly collected frames. The imagination horizon was decreased to 10.

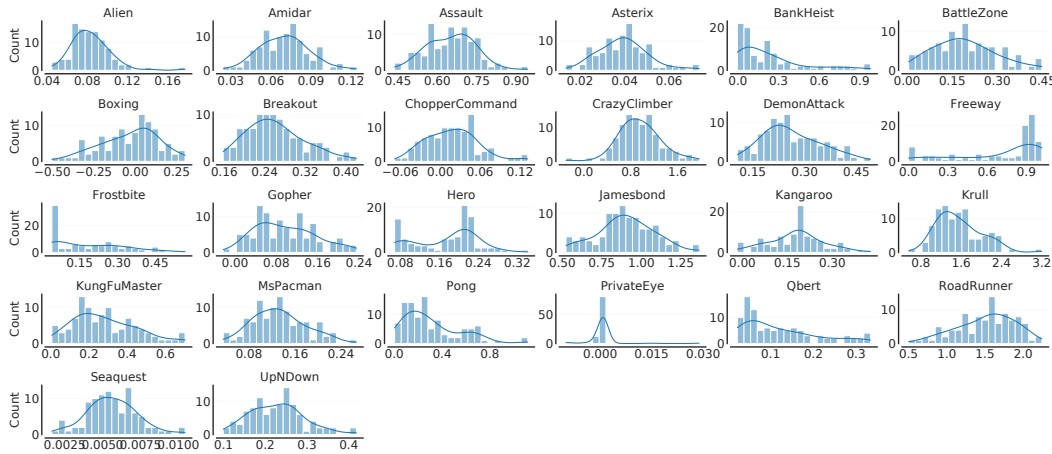

Figure A.14: **Per-game score distributions**. Histogram plot with kernel density estimate of human-normalized scores of DER on 26 games in the Atari 100k benchmark. Each histogram plot is based on 100 runs per game. For most games, the distributions are either skewed (*e.g.,* KUNGFUMASTER), heavy-tailed (*e.g.,* BANKHEIST, FROSTBITE) or multimodal (*e.g.,* HERO)

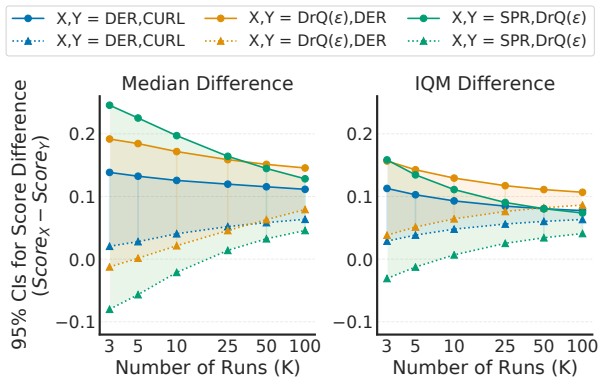

Figure A.15: **Detecting score differences**. **Left**. 95% CIs for differences in median scores. **Right**. 95% CIs for differences in IQM scores. Median requires many more runs than IQM for small uncertainty.

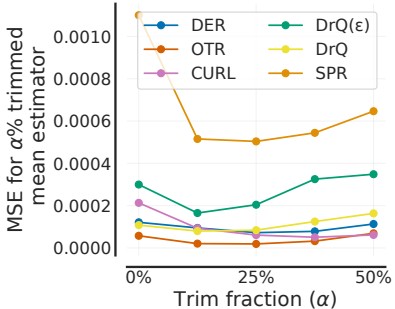

Figure A.16: **Statistical Efficiency of IQM**. Efficiency of an estimator is typically measured in terms of its mean squared error (MSE). We estimate MSE for trimmed estimators with 10 runs by subsampling 20,000 sets of 10 runs with replacement from 100 runs.

**Compute**. For the case study on Atari 100k, we used Tesla P100 GPUs for all the runs. Each run spanned about 3-5 hours depending on the algorithm, and we ran a total of 100 runs / game $\times$ 26 games/algorithm $\times$ 6 algorithms = 15,600 runs. Additionally, we ran an additional 100 runs per game for DER to compute a good approximation of point estimates for aggregate scores, which increases the total number of runs by 2600. Overall, we trained and evaluated 18,200 runs, which roughly amounts to 2400 days – 3600 days of GPU training.

**Comparing performance of two algorithms**. When confidence intervals (CIs) overlap for two random variables $X$ and $Y$ overlap, we estimate the 95% CIs for $X - Y$ to account for uncertainty in their difference (Figure A.15). For example, when using 5 runs, the median score improvement from DrQ($\varepsilon$) over DER is estimated to lie within $(0.01, 0.21)$ while that of SPR over DrQ lies within $(-0.09, 0.18)$. Furthermore, while improvement from SPR over DER with 5 to 15 runs is not statistically significant, claiming "no improvement" would be misleading as evaluating more runs indeed shows that the improvement is significant.

**Analyzing efficiency and bias of IQM**. Theoretically, trimmed means, are known to have higher statistical efficiency for mixed distributions and heavy-tailed distributions (Cauchy distribution), at the cost of lower efficiency for some other less heavily-tailed distributions (normal distribution) than mean, as shown by the seminal work of Tukey [106]. Empirically, on Atari 100k, IQM provides good statistically efficiency among trimmed estimators across different algorithms (Figure A.16) as well as has considerably small bias than median (*c.f.* Figure A.17 *vs.* Figure 3).

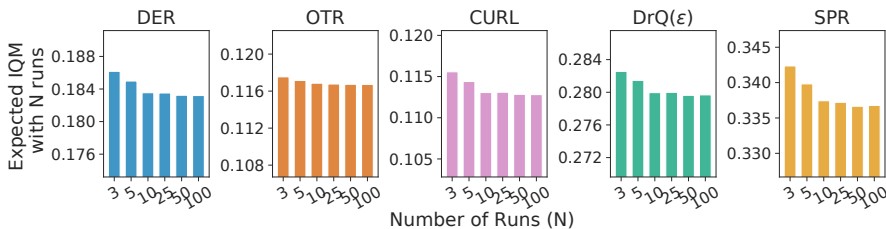

Figure A.17: **Negligible bias in IQM scores**. Expected IQM scores with varying number of runs. The expected score for $N$ runs is computed by repeatedly subsampling $N$ runs with replacement out of 100 runs for 100,000 times. Compared to expected median score differences (Figure 3), the difference in expected IQM scores with 3 runs and 100 runs is typically an order of magnitude smaller. For example, the expected median differences for SPR is 0.05 points while expected IQM differences are only 0.006 points.

### A.3 Related work on rigorous evaluation in deep RL

While prior work [41, 46, 68] highlights various reproducibility issues in policy-gradient methods, this paper focuses specifically on the reliability of evaluation procedures on RL benchmarks and provides an extensive analysis on common deep RL algorithms on widely-used benchmarks.

For more rigorous performance comparisons on a single RL task, Colas et al. [21], Henderson et al. [41] provide guidelines for statistical significance testing while Colas et al. [20] focuses on determining the minimum number of runs needed for such comparisons to be statistically significant. Instead, this paper focuses on reliable comparisons on a suite of tasks and mainly recommends reporting stratified bootstrap CIs due to the dichotomous nature and wide misinterpretation of statistical significance tests (see Remark in Section 2). Colas et al. [20, 21], Henderson et al. [41] also discuss bootstrap CIs but for reporting single task mean scores – however, 3-5 runs is a small sample size for bootstrapping: on Atari 100k, for achieving true coverage close to 95%, such CIs require at least 20-30 runs per task (Figure A.18) as opposed to 5-10 runs for stratified bootstrap CIs for aggregate metrics like median, mean and IQM (Figure A.19).

Chan et al. [16] propose metrics to measure the reliability of RL algorithms in terms of their stability during training and their variability and risk in returns across multiple episodes. While this paper focuses on reliability of evaluation itself, performance profiles showing the tail distribution of episodic returns, applicable for even a single task with multiple runs, can be useful for measuring reliability of an algorithm's performance.

Jordan et al. [49] propose a game-theoretic evaluation procedure for "complete" algorithms that do not require any hyperparameter tuning and recommend evaluating between 1,000 to 10,000 runs per task to detect statistically significant results. Instead, this work focuses on reliably evaluating performance obtained after the hyperparameter tuning phase, even with just a handful of runs. That said, run-score distributions based on runs with different hyperparameter configurations might reveal sensitivity to hyperparameter tuning.

An alternative to *score distributions*, proposed by Recht [83], is to replace scores in a performance profile [26] by the probability that average task scores of a given method outperforms the best method (among a given set of methods), computed using the Welsh's t-test [113]. However, this profile is (1) also a biased estimate, (2) less robust to outlier runs, (3) is insensitive to the size of performance differences, *i.e.,* two methods that are uniformly 1% and 100% worse than the best method are assigned the same probability, (4) is only sensible when task score distributions are Gaussian, as required by Welsh's t-test, and finally, (5) the ranking of methods depends on the specific set of methods being compared in such profiles.

### A.4 Non-standard Evaluation Protocols Involving Maximum

Even when adequate number of runs are used, the use of non-standard evaluation protocols can result in misleading comparisons. Such protocols commonly involve the insertion of a maximum operation inside evaluation, *across* or *within* runs, leading to a positive bias in reported scores compared to the standard approach without the maximum.

One seemingly reasonable but faulty argument [10] for maximum across runs is that in a real-world application, one might wish to run an stochastic algorithm $A$ for $N$ runs and then select the best result. However, in this case, we are not discussing $A$ but another algorithm $A^N$, which evaluates $N$

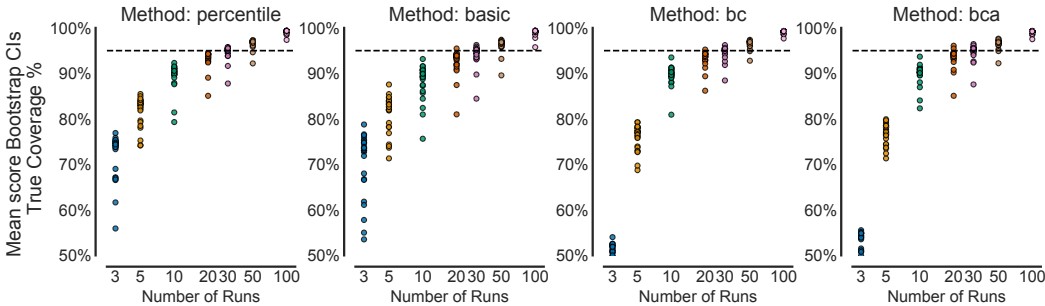

Figure A.18: **Validating 95% bootstrap CIs for per-game mean scores** for a varying number of runs for DER, shown as a scatter plot where each point corresponds to one of the 26 games in Atari 100k. For a given game, the true coverage % is computed by sampling 10,000 sets of K runs without replacement from 200 runs and checking the fraction of 95% CIs that contains the true mean score for that game based on 200 runs. For many games, the true coverage for per-game CIs is below the nominal coverage of 95% even with 30 runs per game.

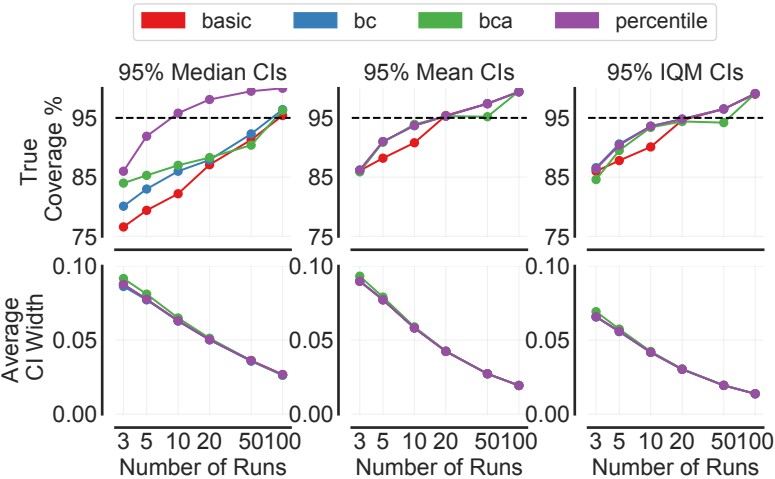

Figure A.19: **Validating 95% stratified bootstrap CIs for aggregate scores** for a varying number of runs. We show CIs for median, mean and IQM scores, aggregated using scores across 26 games, for DER. The true coverage % is computed by sampling 10,000 sets of K runs without replacement from 200 runs and checking the fraction of 95% CIs that contains the true estimate approximation based on 200 runs. Please note that coverage above 95%, even with 50+ runs, is likely due to approximation error in the true estimate using finite runs.

random runs of $A$. If we are interested in $A^N$, taking maximum over $N$ runs only considers a single run of $A^N$. Since $A^N$ is itself stochastic, proper experimental methodology requires multiple runs of $A^N$. Furthermore, because learning curves are not in general monotonic, results produced under the maximum-during-training protocol are in general incomparable with end-performance reported results. In addition, such protocols introduces an additional source of positive statistical bias, since the maximum of a set of random variables is a biased estimate of their true maximum.

On Atari 100k, CURL [56] and SUNRISE [59] used non-standard evaluation protocols. CURL reported the maximum performance over 10 different evaluations during training. As a result, natural variability in both evaluation itself and in the agent's performance during training contribute to overestimation. Applying the same procedure to CURL's baseline DER leads to scores far above those reported for CURL (Figure 5, "DER (CURL's protocol)"). In the case of SUNRISE, the maximum was taken over eight hyperparameter configurations separately for each game, with three runs each. We simulate this procedure for DER (also SUNRISE's baseline), using a dummy hyperparameter. We find that a lot of SUNRISE's improvement over DER can be explained by this evaluation scheme (Figure 5, "DER (SUNRISE's protocol)").

## A.5  Bootstrap Confidence Intervals

Bootstrap CIs for a real parameter $\theta$ are based on re-sampling with replacement from a fixed set of $K$ samples to create a bootstrap sample of size $K$ and compute the bootstrap parameter $\theta^*$ and repeating

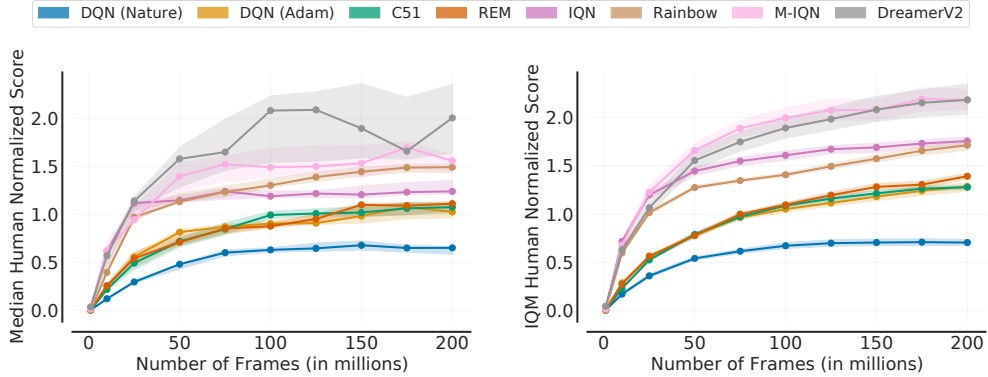

Figure A.20: **Comparing Median vs IQM on Atari 200M**. Sample-efficiency of agents as a function of number of frames measured via median (**left**) and IQM (**right**) human-normalized scores. Shaded regions show pointwise 95% percentile stratified bootstrap CIs. IQM results in significantly smaller CIs than median scores.

this process a numerous to create the bootstrap distribution over $\theta^*$. In this paper, we evaluate the following non-parametric methods for constructing CIs for $\theta$ using this bootstrap distribution:

1. **Basic** bootstrap, also known as the reverse percentile interval, uses the empirical quantiles from the bootstrap distribution of the parameter $\widehat{\delta} = \widehat{\theta} - \theta$ for defining the $\alpha \times 100\%$ CI: $(2\widehat{\theta} - \theta^*_{(\alpha/2)}, 2\widehat{\theta} - \theta^*_{(1-\alpha/2)})$, where $\theta^*_{(1-\alpha/2)}$ denotes the $1 - \alpha/2$ percentile of the bootstrapped parameters $\theta^*$ and $\widehat{\theta}$ is the empirical estimate of the parameter based on finite samples.

2. **Percentile** bootstrap. The percentile bootstrap proceeds in a similar way to the basic bootstrap, using percentiles of the bootstrap distribution, but with a different formula: $(\theta^*_{(1-\alpha/2)}, \theta^*_{(\alpha/2)})$ for defining the $\alpha \times 100\%$ CI.

3. **Bias-corrected** (bc) bootstrap – adjusts for bias in the bootstrap distribution.

4. **Bias-corrected and accelerated** (bca) bootstrap, by Efron [29], adjusts for both bias and skewness in the bootstrap distribution. This approach is typically considered to be more accurate and has better asymptotic properties. However, we find that it is not as effective as percentile methods in the few-run deep RL regime.

More technical details about bootstrap CIs can be found in [40]. We find that bootstrap CIs for mean scores per game (computed using $N$ random samples) require many more runs than aggregate scores (computed using $MN$ random samples) for achieving true coverage close to the nominal coverage of 95% (*c.f.* Figure A.18 *vs.* Figure A.19).

**Number of bootstrap re-samples**. Unless specified otherwise, for computing uncertainty estimates using stratified bootstrap, we use 50,000 samples for aggregate metrics and 2000 samples for pointwise confidence bands and average probability of improvement. Using larger number of samples then the above specified values might result in more accurate uncertainty estimates but would be slower to compute.

**Stratified bootstrap over tasks and runs**[12]. With access to only 1-2 runs per task, stratified bootstrapping can be done over tasks (Figure A.22), to answer the question: "If I repeat the experiment with a different set of tasks, what performance an algorithm is I expected to get?" It shows the sensitivity of the aggregate score to a given task and can also be viewed as an estimate of performance if we had used a larger unknown population of tasks [*e.g.*, 90, 94]. Compared to the interval estimates in Figure 9, bootstraping over tasks results in much larger uncertainty due to high variations in performance across different tasks (*e.g.,* easy vs hard exploration tasks).

## A.6 Visualizing score distributions

**Choice of Normalization**. We used existing normalization schemes which are prevalent on benchmarks including human normalized scores for Atari 100k and ALE, PPO normalized scores and Min-Max normalized scores for Procgen, and Min-Max Normalized scores (minimum scores set

---

[12]Thanks to David Silver and Tom Schaul for suggesting stratified bootstrapping over tasks.

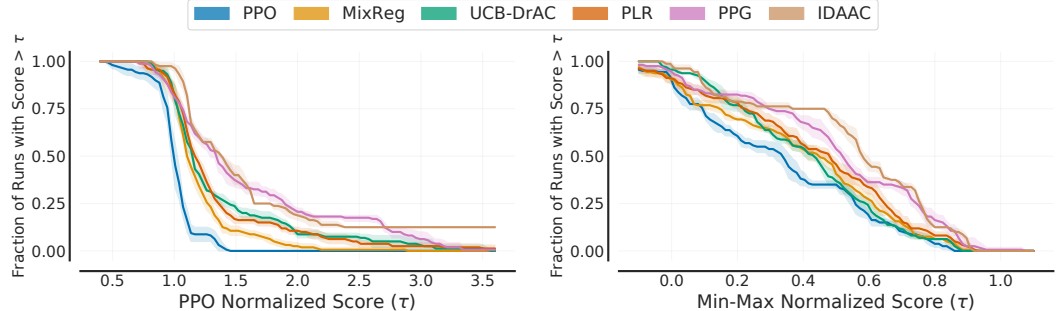

Figure A.21: **Score Distributions on the Procgen benchmark** [18] based on results in the easy mode setting [80]. Shaded regions indicate 95% CIs estimated using the percentile bootstrap with stratified sampling. We compare PPO [92], MixReg [111], UCB-DrAC [81], PLR [48], PPG [19] and IDAAC [80]. We recommend using min-max normalized scores as opposed to PPO normalized scores.

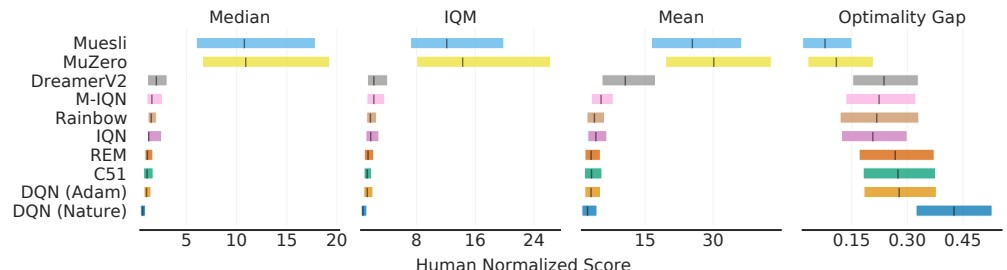

Figure A.22: **Stratified Bootstrap across tasks and runs**. Aggregate metrics on Atari 200M with 95% CIs based on 55 games with sticky actions [69]. Higher mean, median and IQM scores and lower optimality gap are better. The CIs are estimated using the percentile bootstrap with stratified sampling across tasks and runs. MuZero [91] results use 1 run/game while Muesli [43] uses 2 runs/game, as provided by the corresponding authors. All other results are based on 5 runs per game except for M-IQN and DreamerV2 which report results with 3 and 11 runs. These estimates are much wider than that obtained via bootstrap over runs (Figure 9).

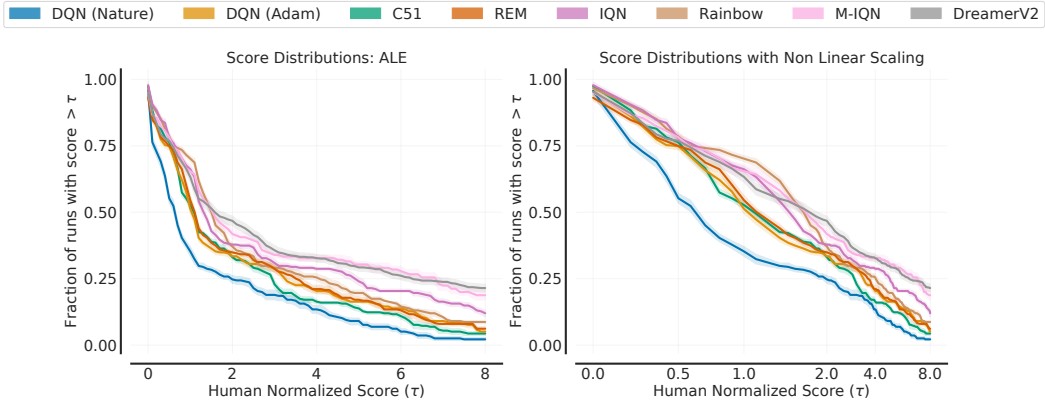

Figure A.23: **Score distributions with linear and with non-linear scaling** on Atari 200M. In the plots above, the x-axis is scaled such that spacing between any two $\tau$ values, $\tau_1$ and $\tau_2$, is proportional to the fraction of runs averaged across algorithms between those two $\tau$ values. This scaling shows the regions of the score distribution where most of the runs lie as opposed to comparing tail ends of the distribution. However, this scaling implies sub-linear utility of achieving higher scores, which may not be accurate as the utility depends on the difficulty of obtaining higher scores – it is much higher to obtain higher scores on hard exploration games. Furthermore, we cannot visually inspect mean/IQM scores based on the area under the curve due to the non-linear scaling.

to zero) scores for DM Control. We do not use record normalized scores for ALE (Figure A.27) in the main text as ALE results are reported by evaluating agents for 30 minutes of game-play as opposed to record scores which were obtained using game play spanning numerous hours (*e.g.,* Toromanoff et al. [105] recommend evaluating agents for 100 hours). Furthermore, we recommend

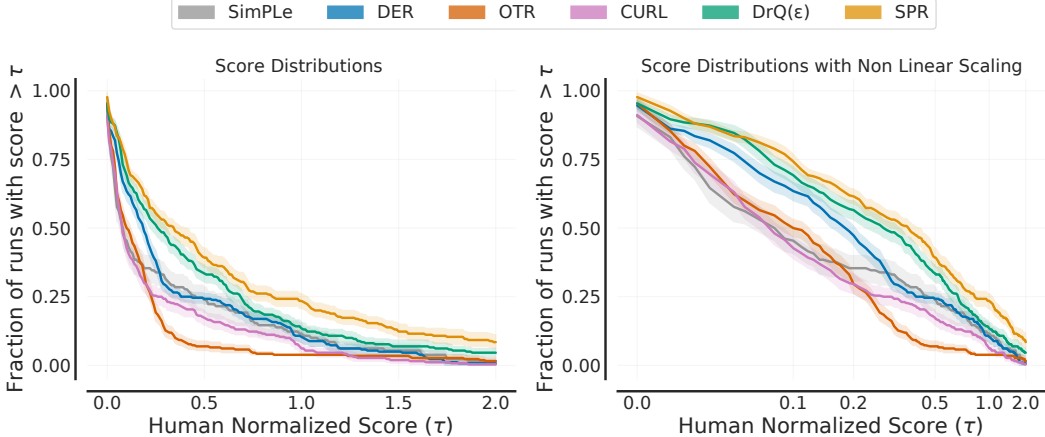

Figure A.24: **Score distributions with linear and with non-linear scaling** on Atari 100k. In the plots above, the x-axis is scaled such that spacing between any two $\tau$ values, $\tau_1$ and $\tau_2$, is proportional to the fraction of runs averaged across algorithms between those two $\tau$ values.

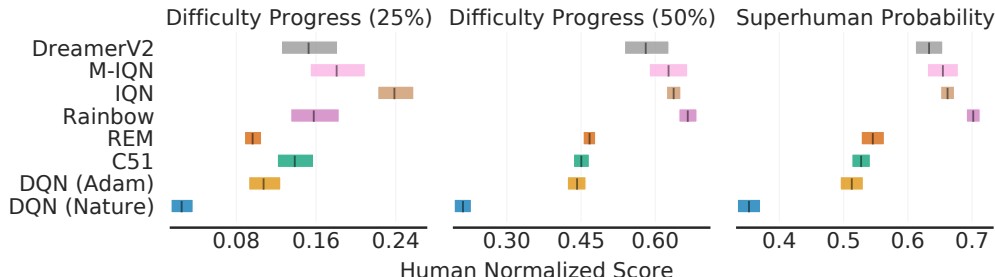

Figure A.25: **Alternative aggregate metrics on ALE** based on 55 games with 95% CIs. Higher metrics are better. The CIs are estimated using the percentile bootstrap with stratified sampling.

using Min-Max Normalized scores for Procgen instead of PPO Normalized scores (Figure A.21) to allow for comparisons to methods which do not build upon PPO [92].

**Scaling x-axis in score distributions**. Figure A.23 (right) and Figure A.24 (right) shows an alternative for visualizing score distributions where we simply scale the $x$-axis depending on the fraction of runs in a given region. This scaling more clearly shows the differences in algorithms by focusing on the regions where most of the runs lie[13].

## A.7  Aggregate metrics: Additional visualizations and details

**Alternative aggregate metrics**. Different aggregate metrics emphasize different characteristics and no single metric would be sufficient for evaluating progress. While score distributions provide a full picture of evaluation results, we provide suggestions for alternative aggregate metrics to highlight other important aspects of performance across different tasks and runs.

- **Difficulty Progress**: One might be more interested in evaluating progress on the hardest tasks on a benchmark [3]. In addition to optimality gap which emphasizes all tasks below a certain performance level, a possible aggregate measure to consider is the mean scores of the bottom 25% of the runs (Figure A.25, left), which we call *Difficulty Progress* (DP-25).

- **Superhuman Probability**: We also recommend reporting *probability of being superhuman*, $P(X > 1)$, given by the number of runs above average human performance (Figure A.25, right) instead of number of games above average human performance [42, 93], a commonly used metric on ALE.

---

[13]Thanks to Mateo Hessel for suggesting this visualization scheme and the difficulty progress metric.

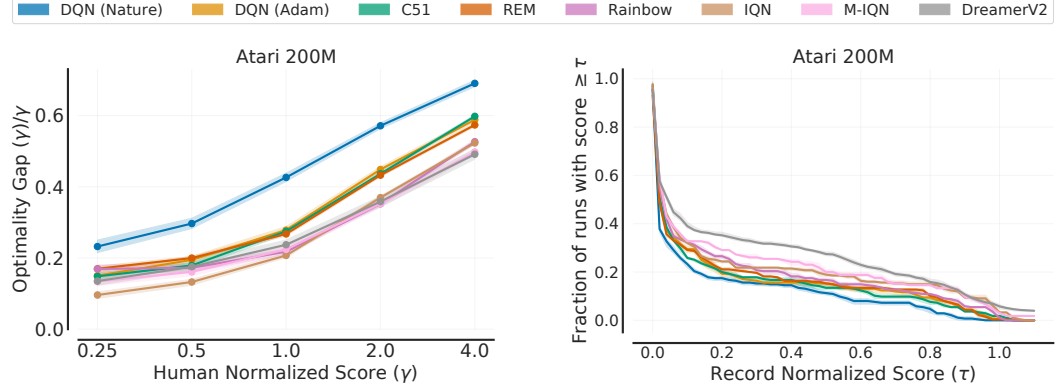

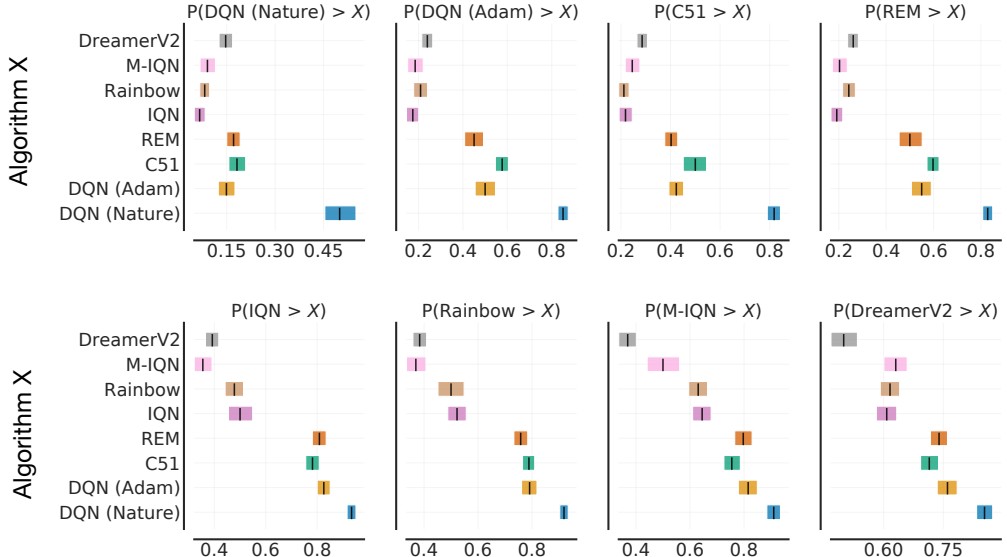

Figure A.26: Optimality gap ($\gamma$) divided by $\gamma$ as a function of $\gamma$. Lower curves are better.

Figure A.27: Score distributions using record normalized scores.

Figure A.28: **Average Probability of Improvement on ALE**. Each subplot shows the probability of improvement of a given algorithm compared to all other algorithms. The interval estimates are based on stratified bootstrap with independent sampling with 2000 bootstrap re-samples

**Choice of $\gamma$ for optimality gap**. When using min-max normalized scores or human-normalized scores, setting a score threshold of $\gamma = 1$ is sensible as it considers performance on games below maximum performance or human performance respectively. If there is no preference for a specific threshold, an alternative is to consider a curve of optimality gap as the threshold is varied, as shown in Figure A.26, which shows how far from optimality an algorithm is given any threshold – a small value of optimality gaps for all achievable score thresholds is desirable.

**Probability of improvement**. To compute the probability of improvement for a task $m$ for algorithms $X$ and $Y$ with $N$ and $K$ runs respectively, we use the Mann-Whitney U-statistic [71], that is,

$$P(X_m > Y_m) = \frac{1}{NK} \sum_{i=1}^{N} \sum_{j=1}^{K} S(x_{m,i}, y_{m,j}) \quad \text{where} \quad S(x,y) = \begin{cases} 1, & \text{if } y < x, \\ \frac{1}{2}, & \text{if } y = x, \\ 0, & \text{if } y > x. \end{cases} \quad \text{(A.2)}$$

Please note that if the probability of improvement is higher than 0.5 and the CIs do not contain 0.5, then the results are statistically significant. Furthermore, if the upper CI is higher than a threshold of 0.75, then the results are said to be statistically meaningful as per the Neyman-Pearson statistical testing criterion by Bouthillier et al. [12]. We show the average probability of improvement metrics

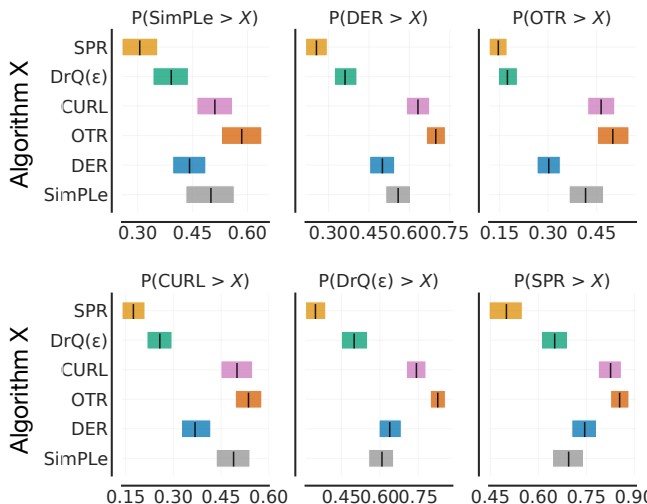

Figure A.29: **Average Probability of Improvement on Atari 100k**. Each subplot shows the probability of improvement of a given algorithm compared to all other algorithms. The interval estimates are based on stratified bootstrap with independent sampling with 2000 bootstrap re-samples.

for Atari 100k and ALE in Figure A.29 and Figure A.28. These estimates show how likely an algorithm improves upon another algorithm.

**Aggregate metrics** on Atari 100k, Procgen and DM Control as well as ranking on individual tasks on DM Control are visualized in Figures A.30–A.33.

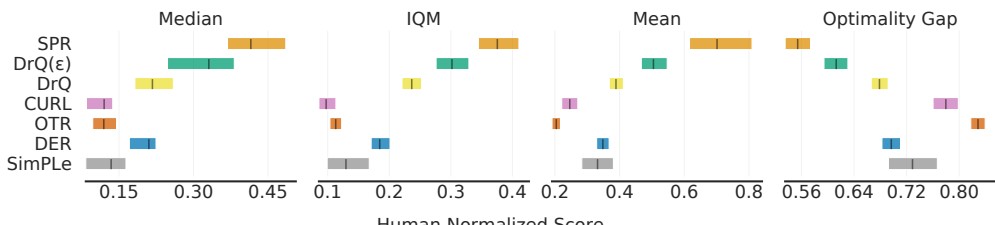

Figure A.30: **Aggregate metrics on Atari 100k** based on 26 games with 95% CIs. Higher mean, median and IQM scores and lower optimality gap are better. The CIs are estimated using the percentile bootstrap with stratified sampling. All results are based on **10 runs per game** except SimPLe, for which we use the 5 runs from their reported results. IQM results in smaller CIs than median scores while optimality gap results in smaller CIs than mean scores. Mean scores are higher than IQM and median scores, indicating that they might be dominated by performance on outlier tasks.

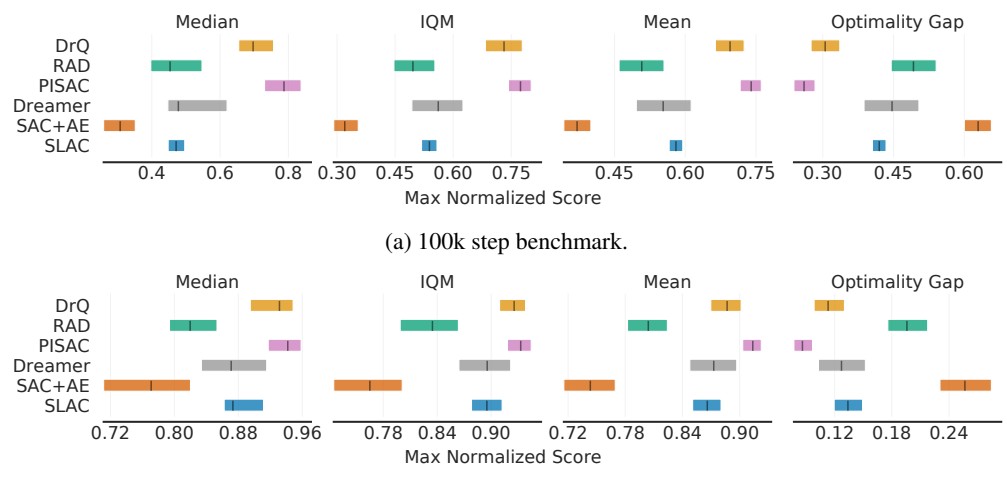

(a) 100k step benchmark.

(b) 500k step benchmark.

Figure A.31: **Aggregate metrics on DM Control** based on 6 tasks with 95% CIs. Higher mean, median and IQM scores and lower optimality gap are better. The CIs are estimated using the percentile bootstrap with stratified sampling with 50,000 bootstrap resamples. All results are based on 10 runs per game. All scores are bounded above by 1, so 1 - optimality gap corresponds to mean scores.

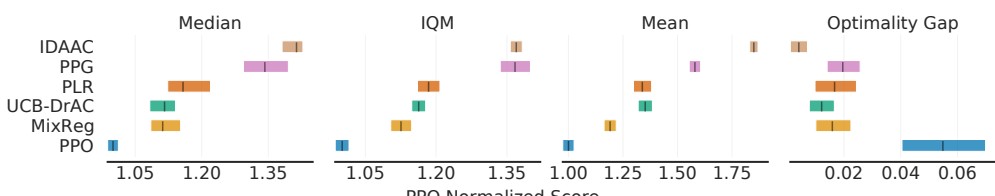

(a) Aggregate metrics based on **PPO normalized scores**. Mean is dominated by outliers while median has large CIs compared to IQM. All algorithms perform better than PPO, resulting in a small optimality gap.

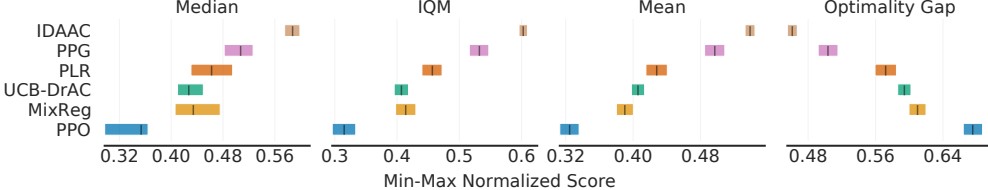

(b) Aggregate metrics based on **min-max normalized scores**. IQM results in smaller CIs than median scores. With min-max normalization, scores are below 1, so optimality gap corresponds to 1 - mean scores.

Figure A.32: **Aggregate metrics on Procgen** based on 16 tasks with 95% CIs. Higher mean, median and IQM scores and lower optimality gap are better. The CIs are estimated using the percentile bootstrap with stratified sampling with 50,000 bootstrap resamples. We compare PPO [92], MixReg [111], UCB-DrAC [81], PLR [48], PPG [19] and IDAAC [80]. All results are based on 10 runs per game.

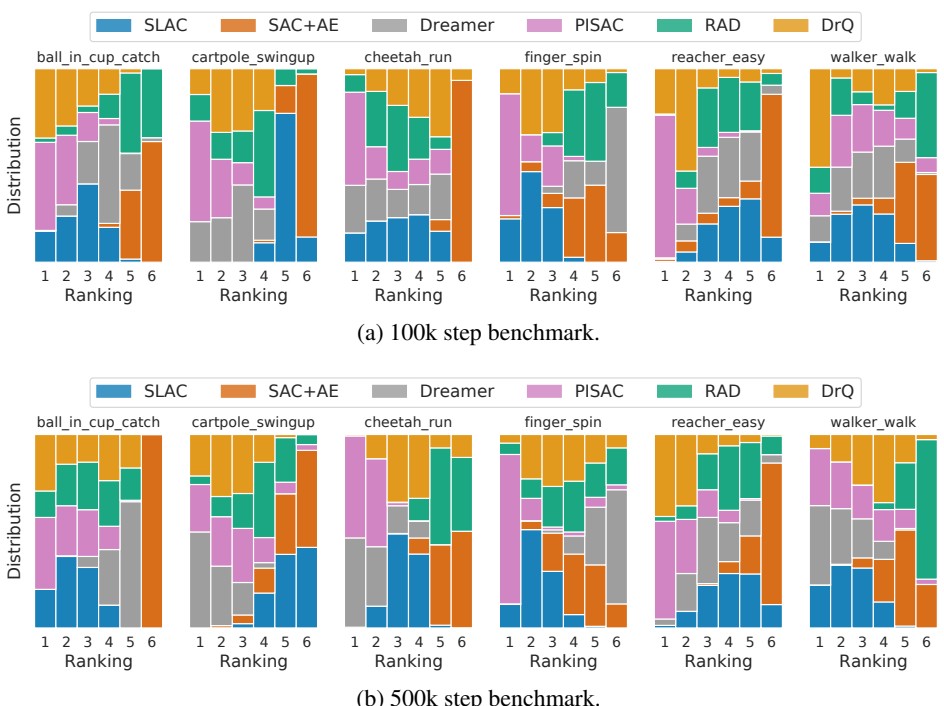

(a) 100k step benchmark.

(b) 500k step benchmark.

Figure A.33: **Ranking on individual tasks on DM Control** 100k and 500k step benchmark. The $i^{th}$ column in the rank distribution plots show the probability that a given method is assigned rank $i$, when compared to other methods. These distributions are estimated using stratified bootstrap with 200,000 repetitions. We observe that no single algorithm consistently ranks above other algorithms on all tasks, making comparisons difficult without aggregating results across tasks.