# OpenReview forum: "Deep Reinforcement Learning at the Edge of the Statistical Precipice"
_NeurIPS.cc/2021/Conference — NeurIPS 2021 Oral_

### Official Review · Reviewer_9Qic · 2021-06-30

**Rating:** 9
**Confidence:** 3

**Summary:**

This paper discusses the evaluation protocol dominant in deep reinforcment learning (evaluating 3-5 runs across a range of games/environments and reporting point estimates of performance such as mean or median), and describes several severe limitations of that protocol. The protocol ignores the uncertainty present in evaluating performance, especially with so few runs. They demonstrate the various failure modes of this protocol with a case study on the Atari@100k benchmark, showing that the reported results of previous methods are misleading and don't capture the variability possible in evaluation. The paper then proposes several methods for more robust evaluation measures which have tighter confidence intervals, including using inter-quartile mean (IQM) instead of mean or median, calculating interval estimates with bootstrap confidence intervals with stratified sampling, and introducing run-score distributions and performance profiles. The authors then demonstrate the use of these tools on Atari@200M, OpenAI Procgen and DeepMind Control from pixels, showing that previous evaluations are sometimes misleading or confusing, and don't properly capture the full nature of the differences between methods.

**Limitations And Societal Impact:**

I believe limitations and societal impacts are adaquately addressed in the paper.

**Main Review:**

## Overview

I believe this paper is an important contribution to the field of RL. It points out a failure mode common in reinforcment learning evaluation and comparison. While similar failures of evaluation have been reported before, it seems (as demonstrated by the case studies in the paper) that these failures haven't been adaquately addressed. Further, the degree of detail and rigour taken in presenting the experimental results (and visualising those results) is a credit to the authors and means the work has an excellent clarity in it's presentation. There are a few small clarifications or adjustments to the paper which I believe would be beneficial.

## Strengths

Originality: As noted in the paper (e.g. L318) there has been previous work looking at the evaluation protocol of deep reinforcement learning, but this paper evaluates these problems more thoroughly, finds problems in previous recommendations of increasing the number of random seeds, and proposes new methods for tackling these issues in deep reinforcement learning evaluation. As noted in the Weaknesses section below, I do think a better comparison of the results, recommendations and methods in this work to previous work would be beneficial to clarify the novelty of the contribution.

Clarity: The paper is very well-written, and explains it's points clearly in all parts. The visualisations are well explained and provide good intuition for the variety of statistical ideas presented in the text of the paper, in a way that is rigorous. The paper's structure is well-organised the motivates the main messages of the paper well.

Quality: The technical quality of the experiments and visualisations is very high, and the techniques used and choices made are well-motivated. The claims made are well supported by the experiments.

Significance: The results presented in the paper of very high significance to the reinforcement learning community. The demonstration of spurious or misleading reporting of previous results is very important and should inform the field as to which methods are actually improvements on previous methods, and in what way. The improved approaches for reporting results, including representing the full distribution of results in multiple ways, should be taken up by current researchers in RL to improve the robustness of their results and determining whether real progress is being made.

## Weaknesses

It would be useful to have a (small) related work section (possibly in the appendix, given the space constraints) on previous work and what it demonstrated, to clarify the contributions in this paper. For example, the papers listed in L318 could be explained in more detail.

### Suggestions for improvements

- L186: I think it would be more fair to say that the increase in the number of runs which would be required to address the statistical uncertainty issues is infeasible for computationally demanding deep RL benchmarks.
- L233: I think the "(1)" should go after "... is also" for this list to read correctly.
- From my understanding of dopamine, using the Rainbow implementation from dopamine isn't the full Rainbow algorithm (it misses out on three components - dueling DQN, double DQN and NoisyNets) - this should be made clear in the paper, as it changes the results on the Atari@200M benchmark. For example, see the Rainbow results in table 2 of [Muesli: Combining Improvements in Policy Optimization](https://arxiv.org/abs/2104.06159), which has a much higher median than reported in your paper (and in the DreamerV2 paper, which also uses the dopamine implementation of Rainbow).

## Edit

Upon reading the response and other reviews, I have raised my confidence from 3 to 4, maintaining my score of 9.

**Time Spent Reviewing:**

3

---

> ### Author Response · Authors · 2021-08-09
> **Author response to reviewer 9Qic**
>
> We thank the reviewer for their valuable feedback and suggestions for improvement. To further clarify the contributions of this paper, we'll add a related work section in the appendix, as recommended by the reviewer. We discuss the papers listed in L318 thoroughly in the common response above.  Our responses follow:
>
> > **Using the Rainbow implementation from dopamine isn't the full Rainbow algorithm (it misses out on three components - dueling DQN, double DQN and NoisyNets) - this should be made clear in the paper, as it changes the results on the Atari@200M benchmark.**
>
> That’s a good point and we’ll mention it. Besides the use of dopamine Rainbow vs full Rainbow, the difference in full Rainbow results and this paper’s results also stem from the fact the original Rainbow paper did not use sticky actions and reported scores in that paper are based on a single run.
>
> > **L186: I think it would be more fair to say that the increase in the number of runs which would be required to address the statistical uncertainty issues is infeasible for computationally demanding deep RL benchmarks.**
>
> Indeed, that’s a better way of phrasing it.
>
> > **L233: I think the "(1)" should go after "... is also" for this list to read correctly.**
>
> Thanks, will fix.

---

> > ### Comment · Reviewer_9Qic · 2021-08-19
> > **Response**
> >
> > Thanks for accepting my suggestions and clarifications. I don't plan to change my score given it is already high, but I'm happy to see that the paper will be improved through the suggestions made here and by the other reviewers, so I will increase my confidence from 3 to 4.

---

> > > ### Author Response · Authors · 2021-08-19
> > > **Thanks!**
> > >
> > > We thank the reviewer again for their suggestions and clarifications. We kindly request the reviewer to raise their confidence in the main review, so that it's clearly visible to the AC.

---

### Official Review · Reviewer_GYfs · 2021-07-11

**Rating:** 7
**Confidence:** 4

**Summary:**

This paper presents an empirical analysis of the failures of evaluation protocols in deep RL research and proposes reporting interval estimates, the use of performance profiles, and the use of more robust aggregate metrics. The empirical analysis focuses on the Atari 100k benchmark (but also looks at Atari 200M, procgen, and the deepmind control suite) and finds that previously reported median scores are not very robust by using more compute to estimate the distribution of normalized scores. The ordering of various algorithms is often different depending on the precise evaluation protocol. The paper proposes bootstrapped interval estimates and performance profiles as better ways to report results that show the variability across runs, and proposes the interquartile mean, optimality gap and probability of improvement as alternatives to the median as aggregate metrics.


**Limitations And Societal Impact:**

The paper discusses societal impact in the appendix. The discussion acknowledges the complexity of reproducibility, but could be a bit more forthcoming about how the proposed methods can help illustrate issues of variability in results, but do not solve this fundamental issue.

**Main Review:**

### Strengths:

1. The paper raises an important problem in deep RL research. While issues of reproducibility have been raised before, this paper focuses specifically on the robustness of evaluation procedures and dives deeper into the issue. The findings of inconsistencies in the literature on this specific part of the pipeline seems like an important problem for the community to consider.
2. The large empirical evaluation provides solid evidence for the paper's claims. This evaluation shows things like the ordering changing across different protocols, the high variability of prior metrics, and the reduced variability in the proposed metrics. Relative to prior work on reproducibility and evaluation protocols, this paper provides a substantially larger evaluation across many seeds and many benchmark suites in different settings with both finite and continuous action spaces that considers many highly performing algorithms in each benchmark. This large-scale evaluation can be useful for the community beyond this paper since the authors plan to release the data from the training runs.
3. The paper's proposed solutions seem to help reduce the problems raised with prior work. The main idea to report intervals or performance profiles instead of point estimates is a clearly better practice. Then the IQM does indeed seem to reduce variability.
4. The paper provides useful tools for the RL community. The paper plans to open source code in a colab notebook to compute all of the proposed confidence intervals, performance profiles, and metrics. It also plans to release all of the data from the large-scale experiments.

---

### Weaknesses:

1. The organization and writing of the paper sometimes lacks focus making it difficult to follow. I understand that evaluation is a complicated issue and that the paper wants to make sure it covers a lot of different aspects and cases which makes it difficult to organize, but the paper could benefit substantially from a more coherent framing. As it is currently written, the paper reads as a laundry list of issues followed by a laundry list of proposals that each solve some parts of the issues . Something like a table that explicitly shows the issues with each method under consideration could help to provide some high level clarity for the reader of the relationship between the shortcomings of prior evaluation procedures versus the new proposals. It would also be useful to answer questions like: what are the tradeoffs between different methods? Can these be presented in a concise visualization/table or theorem/lemma?
2. The paper lacks a more formal argument for the proposed metrics. Adding some theory about when and how the proposed metrics are more robust could help clarify the contribution. Even some illustrative examples (perhaps in the appendix) could help to show what sorts of failure modes the new methods avoid.

---

### Recommendation:

Accept

This paper raises an important problem for deep RL and presents a large empirical study along with some new proposals like reporting bootstrapped confidence intervals. While the organization and theory in the paper could be improved, I think the contribution is good enough to warrant acceptance.

---

### Typos/minor comments/questions:

1. Why does the paper not compute all of the proposed metrics for each task? For example, there is no performance profile or IQM for the control suite, the probability of improvement seems to only be computed on procgen, and . If all of these metrics are included in the paper, then these results should be reported in the appendix. Alternatively, maybe things would be cleaner if one or two of the metrics were removed from the paper.
2. The clause "we assert reporting interval estimates of aggregate performance" on line 15 does not make sense. Perhaps the authors meant something like "advocate for" instead of "assert" here?
3. This is sort of a nitpick, but I don't think the title is very well-aligned with the content of the paper. It would be useful to at least say something about "evaluation" in the title so that the reader gets an idea of what the paper is about.
4. The proposed metrics often seems to order the metrics differently, eg. IQM vs. optimality gap in Figure 9. Of course this is to be expected since they are measuring different things, but more of a discussion about how to deal with this inconsistency could be beneficial.

**Time Spent Reviewing:**

5

---

> ### Author Response · Authors · 2021-08-09
> **Author response to reviewer GYfs**
>
> We thank the reviewer for their valuable feedback and suggestions. As suggested by the reviewer, we'll add a table that explicitly shows the issues with current evaluation methods and our recommendations to fix those issues. We'd also include examples in the appendix to show the failure modes of proposed metrics avoided by the prevalent metrics. Our responses follow:
>
> > **The organization sometimes lacks focus making it difficult to follow .. evaluation is a complicated issue .. more coherent framing .. table that explicitly shows the issues with each method under consideration could serve to provide some high level clarity for the reader.**
>
> We agree that such a table would make it easier for the reader to understand the main takeaways. As presented in our common response, we’ll add a table containing the desiderata for reliable evaluation, the shortcomings of the current prior evaluation approach and our recommendation to address those shortcomings.
>
> > **.. theory for when and how the proposed metrics are more robust .. Even some illustrative examples could help to show what sorts of failure modes the new methods avoid.**
>
> IQM, which considers the performance on the middle 50% of all the combined runs across tasks, avoids these failure modes of the prevalent metrics:
>  - **Median of task means** doesn’t change even if we set the normalized scores on nearly half of the tasks to be 0 – IQM would indicate a severe degradation instead.
>  - **Mean task performance** is dominated by outlier tasks and does not typically indicate performance of most tasks: IQM would ignore these outlier tasks.
>    - For example, JamesBond on Atari 200M gets a normalized score above 50 and results in extremely high mean for Dreamer/M-IQN.
>
> Theoretically, trimmed means, are known to have higher statistical efficiency for mixed distributions and heavy-tailed distributions (Cauchy distribution), at the cost of lower efficiency for some other less heavily-tailed distributions (normal distribution) than mean, as shown by the seminal work of Tukey [1]. We validate this empirically in Figure A.16 (Appendix) which shows the statistical efficiency of various trimmed estimators for different algorithms.
>
> Compared to mean, **optimality-gap** is not prone to outliers and has lower variance – we will add a figure similar to Figure 4 showing that the minimum runs needed to detect an improvement for optimality-gap is much smaller than mean.
>
> [1] Tukey, John W. "A survey of sampling from contaminated distributions." Contributions to probability and statistics (1960): 448-485.
>
> > **The proposed metrics often order the algorithms differently..  this is expected since they are measuring different things, but more of a discussion about how to deal with this inconsistency could be beneficial.**
>
> The inconsistency across metrics arises from the fact that an aggregate metric captures only a specific aspect of overall performance across various tasks. Since performance profiles capture the full picture, they would illustrate why such inconsistencies exist. For example, optimality gap and IQM can be both read as areas in the profile plot (Figure 8). We’ll remark about this in the paper.
>
> > **Societal impact .. acknowledge the complexity of reproducibility .. could be a bit more forthcoming about how the proposed methods can help illustrate issues of variability in results, but do not solve this issue.**
>
> We agree that ensuring reproducibility is going to take much more than a single paper and this paper only partly addresses it by providing tools for more reliable evaluation. We’ll add this point in the societal impacts.
>
>
> > **.. compute all the proposed metrics for each task?**
>
> We’d include all the proposed metrics in the appendix for all the benchmarks.
>
> > **perhaps the authors meant  "advocate for" instead of "assert"  reporting interval estimates of aggregate performance” ?**
>
> Yes.
>
> > **Nitpick .. useful to at least say something about "evaluation" in the title so that the reader gets an idea of what the paper is about.**
>
> What do you think about "Deep RL at the Edge of Statistical Precipice: Revising Current Evaluation Protocols”  as the title?
>
> -----------------------------------------------------------------------------------------------------------------------------------------------------------------------------------------
> *We would appreciate it if the reviewer can confirm that their concerns had been addressed. We’d be happy to engage in further discussions.*

---

### Official Review · Reviewer_KFpZ · 2021-07-14

**Rating:** 8
**Confidence:** 4

**Summary:**

This paper analyzes a large set of existing deep RL methods, proposes a variety of protocols towards more rigorous evaluation even in the presence of a small count ("handful") of runs, and, through an extensive set of analyses, demonstrates a variety of erroneous conclusions drawn from previous evaluation in prior work. The main protocols proposed are (1) to use confidence intervals via bootstrap estimation, (2) to use so-called "performance profiles" / "run-score distributions" for any given randomized score (scalar performance metric), (3) to use additional metrics, mainly the Interquartile Mean (IQM), but also "optimality gap", "probability of improvement", "difficulty progress", and "probability of being superhuman".

**Ethical Concerns:**

Not applicable.

**Limitations And Societal Impact:**

There was no centralized discussion of limitations. This needs to be included, although I'm not sure what the limitations might be. Perhaps one of the major limitations in the proposed protocol is the unclear incentives future deep RL papers have in actually employing the proposed protocol. I speculate that a lack of rigor in some prior work is partly due to a missing / weak incentive for papers to be more rigorous, as it generally entails more nuanced, tempered claims. I think that it might make sense to explicitly mention this phenomenon as a barrier to the adoption of this work.

The societal impacts aren't really addressed: section A.5 actually discussed 'research community impacts', which is a small subset of 'societal impacts'. Perhaps one sentence is needed to explicitly state that the paper poses no foreseeable strongly negative impacts (if the authors agree), and another sentence that discusses how the paper could positively impact society by constituting a step forwards in rigorous small run count-based evaluation, which reduces computational burden (and is therefore "greener" than using large run counts).

**Main Review:**

Main comments
--
This paper is another step in the growing area of research concerned with establishing a more rigorous evaluation of deep RL algorithms. The paper's main strengths are the originality of the proposed protocols (particularly the performance profile), as well as an extensive performance analysis of a large set of deep RL methods. The paper is of high-quality -- the experiments are illuminating because they are very relevant and well-done, and the writing is overall quite clear. The paper could prove quite significant if its proposals are adopted.

One of the paper's weaknesses is that its conclusions are interspersed throughout the paper and appendix, which is more difficult to digest than a clearer "executive summary" of proposed protocols and new findings. While I think it's likely that most readers may agree with the spirit of more rigorous analysis, I think it's likely that many readers that do not closely read the paper may miss the important recommendations and findings. The abstract mentions "we present a more rigorous evaluation methodology", but the reality of the paper is that the methodological recommendations are scattered throughout, as opposed to consolidated into a single compact proposed methodology. Relatedly, the abstract claims that the paper "reveal[s] discrepancies in prior comparisons", but these are not really consolidated or summarized in one place. These issues could be addressed by either making enumerated lists or tables of both (1) the proposed protocols / methodology (e.g. use CIs, use performance profiles, use IQM), and (2) the new findings that shed more light on previous findings (those described in 5.1, as well as those in the appendix).

Another weakness is the underdeveloped discussion of how this paper relates to prior work (L318 [11,15,16,32,37]). For example, [15,16,32,37] also discuss CIs for deep RL.

Other comments
--
The bars for CURL in Fig2. is very faint and hard to see -- the bars of the other methods are much easier to see.

L96 The collection of scores, $y_{1:K}$, is undefined in terms of $x$.

L165-169 The purpose of this experiment is not clearly explained at the beginning of the paragraph. The paper should describe why this experiment is run; from my understanding, it might be something like "Consider a setting in which an algorithm is known to be better -- what is the reliability of median and IQM scores for accurately assessing performance differences as the true performance difference and number of runs varies?"

Fig 6: The definition of "true coverage %" appears to only exist in the caption?

Looking at [19], there are a few difference between the score distributions proposed herein and the originally proposed performance profiles. These differences should be explicitly named and discussed. Two important differences are (1) performance profiles were defined without any bootstrapping to estimated CIs (2) performance profiles are empirical CDFs, whereas score distributions are their complements (tail distributions).

The argument that median is not a good indicator because it is calculated using "at most two tasks" (L256) is dubious. The median is an aggregate metric that is calculated using 100% of the runs, not 1 or 2.

The "IQM" acronym appears long before it is actually defined on L249.

L254 Is the following formal claim? "IQM has considerably less bias than median". If so, then this claim requires evidence in the form of a derivation or citation, particularly because the audience for this paper is likely not intimately familiar with the IQM.

The ability to read the medians from the performance profiles would be better highlighted if the Fig 7 y=0.5 lines were thicker.

L297 "While publications make binary claims" -> "While publications sometimes make binary claims". The former statement is too broad to be true.

L267 It's not clear how P(X_m > Y_m) is estimated. More details are needed. After looking through the appendix, it appears to be on page 20. This definition should be referenced in the main paper near L267.

Appendix P20, it's not clear if "Alternative alternative Metrics" is a typo or intentional

L323 "... the problem is not solved by fixing random seeds, as has sometimes been proposed" -- consider adding citations here for epistemological completeness.


**Time Spent Reviewing:**

4

---

> ### Author Response · Authors · 2021-08-09
> **Author response to reviewer KFpZ**
>
> We thank the reviewer for their valuable feedback and suggestions. To address the weaknesses pointed out, we’ll include an executive summary and add an related work section for more thorough discussion of prior work. Our responses follow:
>
> > **One of the paper's weaknesses is that its conclusions are interspersed throughout the paper and appendix, which is more difficult to digest than a clearer "executive summary" of proposed protocols and new findings.**
>
> As presented in our common response above, we’ll add a table containing the desiderata for reliable evaluation, the shortcomings of the current prior evaluation approach and our recommendation to address those shortcomings. We’ll also list the new findings which shed light on previous findings in the appendix.
>
> > **Another weakness is the underdeveloped discussion of how this paper relates to prior work .. prior work also discuss CIs for deep RL.**
>
> [15,16,32] also discuss bootstrap CIs but for reporting single task mean scores – however, 3-5 runs is a really small sample size bootstrapping: on Atari 100k, for achieving true coverage close to 95%, such estimated 95% CIs require at least 20-30 runs per task (Figure A.18) as opposed to N=5-10 runs for stratified bootstrap CIs for aggregate metrics like median, mean and IQM.
>
> We discuss other prior work more thoroughly in the common response and will include it in the paper.
>
> > **The argument that median is not a good indicator because it is calculated using "at most two tasks" (L256) is dubious.**
>
> Indeed, this statement was confusingly written. What we intended to say was that median performance only depends on the performance ordering across tasks and not on the magnitude of performance except at most 2 tasks. For example, zero performance on nearly half of the tasks do not affect median performance. We'll clarify this in the revision.
>
> > **I speculate that a lack of rigor in some prior work is partly due to a missing / weak incentive for papers to be more rigorous, as it generally entails more nuanced, tempered claims. I think that it might make sense to explicitly mention this phenomenon as a barrier to the adoption of this work.**
>
> Barriers to adoption of proposed protocols is something we deeply care about too and would incorporate in centralized limitations in the discussion section.  Currently, the main incentive for adoption seems to be about doing good and reproducible science. Furthermore, we hope that our findings about erroneous conclusions in published papers would encourage researchers to avoid fooling themselves, even if that requires nuanced and tempered claims. That said, maybe a more pragmatic incentive would be if conferences and reviewers required more rigorous evaluation for publication (e.g., NeurIPS checklist asks whether error bars are reported).
>
> Moving towards more reliable evaluation is an ongoing process and we believe that this paper would greatly benefit it.
>
> > **Section A.5 actually discussed 'research community impacts', which is a small subset of 'societal impacts' .. explicitly state that the paper poses no foreseeable strongly negative impacts (if the authors agree), and another sentence that discusses how the paper could positively impact society by constituting a step forwards in rigorous small run count-based evaluation, which reduces computational burden (and is therefore "greener" than using large run counts).**
>
> We were also not able to think of any significant negative impacts and eschewed from mentioning it to avoid painting an overly positive picture. However, we totally agree with the reviewer’s points and would add them to the societal impact section.
>
> We thank the reviewer for their other comments about improving the clarity of the paper and will address them in the camera ready.

---

> > ### Comment · Reviewer_KFpZ · 2021-08-16
> > **Reviewer response to authors**
> >
> > Authors, thank you for addressing my concerns. I think that the described changes will improve the paper. I'll maintain my clear accept rating (8) and increase my confidence from 3 to 4. I have another suggestion that may be worth implementing within the executive summary table, if space allows: for each cell, link to the main supporting evidence (tables and figures or subsections). This will strengthen the summary by making it clear that these recommendations are backed by reasoning and evidence presented in the paper, and enable easier inspection of the reasoning and evidence for each individual recommendation. Finally, a minor nitpick (please excuse the pedantry): to match the singular used in the other column titles of the executive summary table, "Desideratum", not "Desiderata", would be correct.

---

### Official Review · Reviewer_9zHZ · 2021-07-16

**Rating:** 7
**Confidence:** 3

**Summary:**

The comparisons between RL algorithms are always based on the mean/median obtained from several runs for each task. Because of the limited number of experiments, the comparison can be overwhelmed by the uncertainty. This paper proposes using new metrics such as IQM to replace mean/median; furthermore, it also argues that reporting performance distributions would be a better choice.


**Limitations And Societal Impact:**

yes

**Main Review:**

Strength:
- This paper has an extensive number of experiments, which is very valuable.
- I appreciate the open-source code, such as colab, that is easy to implement.
- The problem is well-motivated, it is important to find a more robust way to make comparisons when the computation resources are limited.

Weakness:
The paper proposes using robust aggregate statistics (e.g. IQM) with interval estimates and performance distributions as measures to make comparisons among RL tasks. I have three concerns/questions regarding this (I'll mainly consider the case N=5, because usual medium and small labs don't have resources beyond that)
- From Fig2 and Fig7, we can see that the interval estimates and performance distributions between different algorithms may have significant overlaps, e.g., DrQ and SPR in Fig2 right panel for 3/5 runs, then can we claim a winner between these two algorithms? Or should we eventually have to compare the single aggregate statistics in this case?
- The authors argue that IQM is a more robust statistics compared with its surrogates such as median, first I don't think they're more robust to outliers than median since median uses less data points than IQM does, so the chance of median getting affected by outliers is less than IQM. It looks obvious that IQM will have a much smaller variance because it averages across more data points, however mean can have even smaller variance. So I think the point here is really how "strange" the normalized score distribution can be: if it's like a Gaussian distribution, mean will be the best choice; if the other extreme situation, median may be more appropriate. If in the middle, IQM may play its role. But we never know what the true distribution looks like (depending on the algorithm and environment), so there is no guideline for this.
- Generalization: as pointed above, the choices and conclusions are made on Atari games mainly because the authors have $100$ runs of each algorithm in every environment (which we can consider them as the ground truth). However, I don't know how much it can be generalized to other task sets or another set of algorithms.

Minor Remarks:
- I would suggest to clarify the tasks for each plot. If all plots are using all the tasks, should the heteroskedasticity be considered when doing the test?
- l.77: It's not independent of the rest of the tasks because the order matters.
- Fig2: Is there any explanation why SPR with a larger $N$ has a larger variance than DER with a smaller $N$?
- Fig2: Why is Median and IQM different for $N=3$? I think both of them would use the middle value in this case.
- Fig6: I'm confused by the description here, I thought each task has 100 runs and using bootstrap with replacement?
- Fig6: It is interesting that Median seems to do a better job than IQM when the cases are small (N<5)
- l.163: It would be helpful to provide details on how this values are calculated. For example, what is the power that requires 50-100 runs?
- eq(1): It should be $\hat{F}_i(\tau)$ for summation



**Time Spent Reviewing:**

4

---

> ### Author Response · Authors · 2021-08-09
> **Author Response to Reviewer 9zHZ**
>
> We thank the reviewer for their valuable feedback and raising practical concerns. Their review suggests that there might be some misunderstandings, which we clarify below:
>
> > **Why is Median and IQM different for N=3? I think both of them would use the middle value in this case.**
>
> Median corresponds to the sample median over tasks means – for N=3 runs/task and `M` tasks (e.g., M=26 for Atari 100k, 16 for Procgen), the median is computed over `M` task mean scores.
>
> IQM is calculated using the middle 50% of all the runs – for N=3 runs/task and `M` tasks, it is an average of `3M/2` scores (so 39 scores for Atari 100k).
>
> > **I suggest clarifying the tasks for each plot. If all plots are using all tasks, should heteroskedasticity be considered when doing the test?**
>
> All tasks are used for each plot (26 tasks for Atari 100k, 57 for Atari 200M, 16 for Procgen and 6 for DM Control).  Following Amrhein et al. [2], Wasserstein et al. [84], we do not use statistical significance tests due to their dichotomous nature and wide misinterpretation. As such, the proposed statistical tools including stratified bootstrap CIs do not assume homoscedasticity.
>
> > **l.77: Median  not independent of the rest of the tasks because the order matters.**
>
> Indeed, this statement was confusingly written. What we intended to say was that median performance only depends on the performance ordering across tasks and not on the magnitude of performance except at most 2 tasks. For example, zero performance on nearly half of the tasks does not affect the median. We'll clarify this in the revision.
>
>
>
>  ### Response to their main concerns:
>
>
> > **interval estimates and performance distributions .. may have significant overlaps .. then can we claim a winner between algorithms? Or should we have to compare the single aggregate statistics?**
>
> While claiming a winner between algorithms may seem pragmatic, one of the main points of the paper is to avoid claiming a winner when there is not enough evidence as claiming winners has resulted in erroneous conclusions in prior publications (Section 3 and 5). Instead, we advocate for more nuanced claims of performance difference supported by interval estimates and performance profiles.
>
> Also, when interval estimates for a performance metric (e.g, IQM) overlap, proper comparisons of random variables entail reporting interval estimates for differences in that metric to capture how likely are the reported improvements when the experiment is repeated. This is discussed in the Appendix A.2 (also see Figure A.15). We will add an explicit pointer to it in the main paper.
>
> > **IQM is less robust to outliers but smaller variance than median.. but mean can have even smaller variance .. it depends on how "strange" the normalized score distribution can be .. never know what the true distribution looks like, so there is no guideline for this.**
>
> We acknowledge that the best aggregate metric is dependent on underlying normalized score distribution – for example, we illustrate the bias-variance tradeoff in Figure A.16 (Appendix) which shows the statistical efficiency of various trimmed estimators. Also, even for tasks with Gaussian distributions with different variances, mean may be far from optimal as variance of mean can be dominated by a single high variance task.  That said, we recommend IQM for reasons specified below.
>
> **Rationale for IQM**: IQM, which considers the performance on the middle 50% of the combined runs across all tasks, avoids the failure modes of the prevalent metrics:
>  - Median task performance doesn’t change even if we set the scores on nearly half of the tasks to be 0 – IQM would indicate a severe degradation instead.
> 	- While median is more robust to outliers than IQM, this robustness comes at the expense of statistical efficiency, which is crucial in the few-run regime: median requires much more runs to claim improvements than IQM.
>  - Mean task performance is dominated by outlier tasks and does not typically indicate performance of most tasks: IQM would ignore these outlier tasks.
>    - For example, JamesBond on Atari 200M gets a normalized score above 50 and results in extremely high mean for Dreamer/M-IQN.
>
> Finally, IQM may have failure modes too, albeit less likely than median and mean – this is why we  advocate for reporting performance profiles that estimate the underlying run-score distribution, which could help in identifying these failures, and reporting other metrics such as optimality gap and probability of improvement to present results in a more complete fashion.
>
> > **Generalization: .. choices and conclusions made on Atari 100k .. how much it can be generalized to other task sets or another set of algorithms.**
>
> We empirically address the generalization question by applying the proposed statistical tools on DM Control, Atari 200M and Procgen on actor-critic, policy gradient, deep Q-learning and model-based RL algorithms (Section 5).
>
> Our proposals including stratified bootstrap CIs for interval estimates and run-score distributions to show performance variability across tasks are xxgeneral and applicable to any RL benchmark with multiple tasks. The generality of our recommendations are  empirically validated on benchmarks other than Atari 100k, for example:
>  - **ProcGen**: Figure A.16 show that IQM has much smaller CIs than median on Procgen and Figure 11 (left) shows that PPO-normalized mean scores are dominated by outliers for recent algorithms.
>  - **Atari 200M**: Similar to Procgen, Figure 9 shows the efficacy of IQM over median and mean.
>
>
> --------------------------------------------------------------------------------------------------------------------------------------------------------------------------------------
>
>
>  #### *Other Minor Remarks*:
>
> > **Fig2: Is there any explanation why SPR with a larger N has a larger variance than DER with a smaller N?**
>
> The reason is that the distribution of task mean score for the median tasks for SPR has a larger variance (even with N=10 runs/task) than that of mean score distributions of median tasks for DER (which are different from SPR’s median tasks). Also, SPR may be an intrinsically higher variance algorithm than DER.
>
> > **l.163: It would be useful to provide details on how these values are calculated. For example, what is the power that requires 50-100 runs?**
>
> The main point here was that the minimum number of runs to claim statistically defensible improvements when using median is around 50-100 as can be gleaned from CIs in Figure 2 (right), Figure A.16 (SPR vs DrQ) and Figure 4. The mention of statistical power diverts the reader's focus away from this finding and we’ll omit it in the revision.
>
> Figure 4 actually shows something closer to a power analysis -- for a given improvement size, what is the uncertainty in the measured improvement size based on median/IQM. While analytic computation of power can be done, it additionally requires setting up the null and alternative hypothesis, a(dichotomous) significance testing procedure and specifying the effect size.
>
> > **Fig6: It is interesting that Median seems to do a better job than IQM when the cases are small (N<5)**
>
> While median may be better in terms of CI coverage for N < 5, it is almost twice as worse in terms of CI width which matters a lot to claim improvements in the few-run regime.
>
> > **Fig6: I'm confused by the description here, I thought each task has 100 runs and using bootstrap with replacement?**
>
> - For the coverage experiment, we ran additional 100 runs for DER for computing a more precise point estimate (using 200 runs) as estimating the true coverage requires checking whether the “true” point estimate lies within the estimated CI. We’ll update the caption to clarify this.
> - The bootstrap CIs are computed with replacement but the 20,000 sets of K runs on which bootstrap CIs are computed are subsampled without replacement. Since we don’t encounter runs with exact same values when computing interval estimates, we use subsampling to obtain K different runs. That said, the true coverage % is nearly identical when sampling K runs with or without replacement.
>
> --------------------------------------------------------------------------------------------------------------------------------------------------------------------------------------
> *We would appreciate it if the reviewer can confirm that their concerns had been addressed and, if so, reconsider their assessment. We’d be happy to engage in further discussions.*

---

> > ### Comment · Reviewer_9zHZ · 2021-08-17
> > **Update**
> >
> > - Thank you for the explanations. For major point 1, I think it's a trade-off between false negative rate and false positive rate when deciding a clear winner. For major point 2 and 3, I was meant to ask about generalization to continuous control environments. However, I agree that the guidance to discrete control environments itself is already a great contribution.
> > - Thank you for addressing all the minor points, it is much more clear now. It would be great if the final revision can take these into consideration.
> > - I'm happy to reevaluate the paper and raise the score.

---

### Author Response · Authors · 2021-08-09
**Common response and updates for revision**

We thank the reviewers for their valuable feedback! Reviewers found the paper to be well-written, of high-quality and of high significance to the RL community. They also commended the extensive empirical evaluation and found the proposed protocols original and quite useful. Based on reviewers' comments, we’ll make the following changes:

- *An executive summary of proposed protocols* for reliable evaluation. [Reviewer GYfs, KFpZ]

| Desideratum | Current evaluation approach |  Our Recommendation    |
| --------------------------------- | ----------- | --------- |
| Uncertainty in aggregate performance | **Point estimates**: (1) Ignore statistical uncertainty (2) Hinder *results reproducibility* | Interval estimates using **stratified bootstrap confidence intervals** (CIs) |
|Performance variability across tasks and runs| **Tables with task mean scores**: (1) Overwhelming beyond a few tasks, (2) Standard deviations frequently omitted, (3) Incomplete picture for multimodal and heavy-tailed distributions. | **Score distributions** (*performance profiles*): (1) Shows tail distribution of scores, (2) Qualitative comparisons, (3) Read any score percentile from the profile|
|Aggregate metrics for summarizing benchmark performance | **Mean**:  Often dominated by performance on outlier tasks. &nbsp; **Median**: (1) Statistically inefficient (requires a large number of runs to claim improvements),  (2) Poor indicator of overall performance: 0 scores on nearly half the tasks doesn't change it.| **Interquartile Mean (IQM)** across runs: (1) Performance on middle 50% of combined runs, (2) Robust to outliers but more statistically efficient than median. Report other aspects of performance gains: *Probability of improvement*, *Optimality gap* |
||||



 - *Failure modes of prevalent metrics avoided by proposed metrics* [Reviewer GYfs, 9zHZ]


- More discussion of *societal impact* and *limitations* related to barriers of adoption of this work [Reviewer KFpZ, GYfs]


 - *Update sample efficiency curve* on Atari 200M (Figure 10, right) – we mistakenly plotted the median normalized score instead of IQM – IQM actually results in significantly smaller CIs for all methods (see the colab in Appendix A.1).


- More discussion of closely **related work** [Reviewer 9Qic, KFpZ]

While prior work highlights various reproducibility issues in policy-gradient methods [32, 54], this paper focuses specifically on the reliability of evaluation procedures on RL benchmarks and provides an extensive analysis on common deep RL algorithms on prevalent benchmarks. For more rigorous performance comparisons on a single RL task, [16, 32] provide guidelines for statistical significance testing while [15] focuses on determining the minimum number of runs needed for such comparisons to be statistically significant.  Instead, this paper focuses on reliable comparisons on a suite of tasks and mainly recommends reporting stratified bootstrap CIs due to the dichotomous nature and wide misinterpretation of statistical significance tests.

[15,16,32] also discuss bootstrap CIs but for reporting single task mean scores – however, 3-5 runs is a small sample size for bootstrapping: on Atari 100k, for achieving true coverage close to 95%, such CIs require at least 20-30 runs per task (Figure A.18) as opposed to 5-10 runs for stratified bootstrap CIs for aggregate metrics like median, mean and IQM.

[11] proposes metrics to measure the reliability of RL algorithms in terms of their stability during training and their variability and risk in returns across multiple episodes.  While this paper focuses on reliability of evaluation itself, performance profiles showing the tail distribution of episodic returns, applicable for even a single task with multiple runs, can be useful for measuring reliability of an algorithm’s performance.

[37] proposes a game-theoretic evaluation procedure for “complete” algorithms that do not require any hyperparameter tuning (HPT) and recommends evaluating between 1,000 to 10,000 runs per task to detect statistically significant results. Instead, this work focuses on reliably evaluating performance obtained after the HPT phase, even with just a handful of runs. That said, run-score distributions based on runs with different hyperparameter configurations might reveal sensitivity to HPT.

---

### Decision · Program_Chairs · 2021-09-27

**Decision:**

Accept (Oral)

**Comment:**

I thank the authors for their submission and active participation in the discussions. The reviewers unanimously agree that this is a very strong paper with a convincing motivation [9zHZ] and thorough experiments [9zHZ,GYfs,9Qic]. It raises an important issue in deep RL research that could significantly help future work [KFpZ,9Qic]. Furthermore, reviewers appreciate the release of open source code and tools that will help future researchers [9zHZ,GYfs] with their evaluation protocol. Overall, this paper presents an important step in more rigorous evaluation of deep RL research. It identified shortcomings of prior evaluation protocols, questioning some of the conclusions made in recent years, as well as providing clear guidance on how to improve evaluation in the future. I agree with the reviewers and strongly recommend acceptance. I also encourage the authors to use the feedback by the reviewers to further improve their paper, in particular it's clarity.